# OPTIMAL BRAIN RESTORATION FOR JOINT QUANTIZATION AND SPARSIFICATION OF LLMS

**Hang Guo,  Luca Benini,  Yawei Li**
ETH Zürich

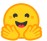   https://huggingface.co/HangGuo/OBR

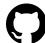   https://github.com/csguoh/OBR

## ABSTRACT

Recent advances in Large Language Model (LLM) compression, such as quantization and pruning, have achieved notable success. However, as these techniques gradually approach their limits, relying on a single method for further compression has become increasingly challenging. In this work, we explore an alternative solution by combining quantization and sparsity. This joint approach, though promising, introduces new difficulties due to the inherently conflicting requirements on weight distributions: quantization favors compact ranges, while pruning benefits from high variance. To attack this problem, we propose Optimal Brain Restoration (OBR), a general and training-free framework that aligns pruning and quantization by error compensation between both. OBR minimizes performance degradation on downstream tasks by building on a second-order Hessian objective, which is then reformulated into a tractable problem through surrogate approximation and ultimately reaches a closed-form solution via group error compensation. Experiments show that OBR incurs only a 1.4 perplexity degradation on Llama2-7B to enable aggressive W4A4KV4 quantization with 50% sparsity, delivering up to **4.72×** speedup and **6.4×** memory reduction compared to the FP16-dense baseline.

## 1 INTRODUCTION

Large Language Models (LLMs) (Brown et al., 2020; Achiam et al., 2023; Dubey et al., 2024) have demonstrated remarkable capabilities across a wide range of tasks. However, as LLMs continue to grow in size with increasing parameter counts, efficiently serving them, especially in resource-constrained edge devices, remains a significant challenge (Dettmers et al., 2022).

To meet the demand for efficient LLM deployment, a variety of methods have been proposed. One prominent line of work focuses on LLM quantization (Nagel et al., 2021), whose main objective is to remove outliers inherent in the LLM weights. To this end, existing works introduce either smoothing (Lin et al., 2024a; Xiao et al., 2023) or Hadamard rotation as a preprocessing step (Ashkboos et al., 2024; Liu et al., 2024) to redistribute outliers before quantization. Thanks to the resulting flat distributions, recent state-of-the-arts (Liu et al., 2024; Sun et al., 2024; Hu et al., 2025) can achieve even 4-bit weight-activation-KV cache (W4A4KV4) inference while preserving performance comparable to the FP16 counterparts. Besides quantization, LLM pruning (Ma et al., 2023; Frantar & Alistarh, 2023) is often considered as another popular solution for compressing LLMs. Recent LLM pruning works (Sun et al., 2023; Zhang et al., 2024) have shown promising results on 50% unstructured and 2:4 semi-structured sparsity by additionally considering the statistics of activations during pruning.

Despite the promising results at moderate compression, relying on a single technique for further reduction is becoming increasingly difficult. As shown in Fig. 1(a), the quantization method QuaRot (Ashkboos et al., 2024) achieves competitive perplexity at moderate bit-width, but suffers from severe degradation under 4-bits. Similarly, pruning alone also faces analogous limitations, where aggressive sparsity inevitably leads to substantial degradation. In this work, we explore an alternative path to go beyond current LLM compressions by jointly leveraging quantization and sparsification. The intuition arises from the observation that low-bit and sparse representations can coexist. We empirically find an average of 14.28% unstructured sparsity in the W4A4KV4 quantization-only Llama2-7B

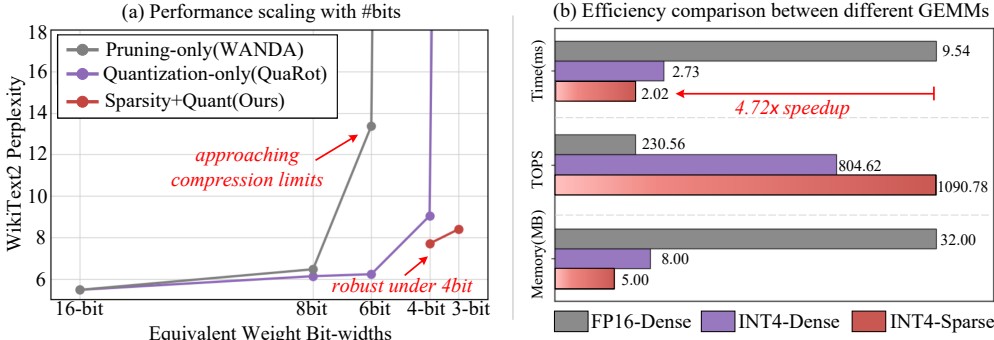

Figure 1: (a) Single compression techniques (Sun et al., 2023; Ashkboos et al., 2024) rapidly reaches limits under sub-4 bits while the joint counterpart can enable further compression. To enable a unified comparison in a single figure, pruning is represented using equivalent bit-widths. (b) INT4 + 2:4 sparse GEMM can achieve faster inference speed, higher throughput, and lower memory usage.

model, suggesting potential combination of quantization and pruning. Furthermore, recent hardware advances, such as NVIDIA's Ampere and Hopper architectures, have introduced native support for INT4-sparse GEMM kernels (Mishra et al., 2021; NVIDIA, 2022; 2021), making the combination of quantization and sparsity increasingly practical for efficient LLM inference.

However, achieving effective joint quantization and sparsification is non-trivial, primarily due to the inherent conflict between their objectives. Specifically, quantization favors a narrow numerical range in the weights to minimize quantization error, whereas pruning benefits from large variations in weight magnitudes to reveal naturally sparse patterns. For instance, Hadamard rotation is a common practice in existing methods to smooth outliers for W4A4KV4 quantization. However, as evidenced by Sec. 5.1, using existing pruning methods to force zeros on the Hadamard-rotated weights leads to unacceptable performance degradation.

**Our approach.** In this work, we propose Optimal Brain Restoration (OBR), a general framework to enable joint quantization and sparsification. The core idea of our OBR is to intervene between pruning and quantization by computing an optimal compensation that minimizes the impact of compression on downstream tasks, thereby reconciling their conflicting requirements on weight distributions. To achieve this, we begin by formulating the second-order Hessian objective to minimize the impact of weight perturbations on downstream task performance. To make the optimization problem tractable, this objective is then approximated through row-wise decoupling, which eliminates inter-row correlations. Building on this surrogate, we further introduce group error compensation, which redistributes distortions from pruning and quantization to minimize overall error, yielding an explainable closed-form solution. By reconciling the conflicting requirements between quantization and sparsity, OBR provides an efficient and practical solution for LLM compression.

To the best of our knowledge, OBR is the first to enable W4A4KV4+50% sparsity LLMs, without requiring any additional retraining. We apply the proposed framework on Llama2 (Touvron et al., 2023), Llama3 (Dubey et al., 2024), and Qwen2.5 (QwenTeam, 2024) families, and demonstrate promising performance with OBR. In particular, our highly compressed model narrows the perplexity gap to merely 1.37 *wrt* its full-precision Llama2-70B counterpart. Furthermore, we evaluate the inference efficiency using INT4 sparse GEMM kernels. As shown in Fig. 1, OBR achieves up to $4.72\times$ speedup and $6.4\times$ memory reduction compared to FP16-dense baselines. We hope our work can serve as a solid baseline and stimulate further research towards sparse low-bit LLMs.

## 2 RELATED WORK

**Network Quantization for LLMs.** Network quantization aims to accelerate inference by converting the full-precision representations into low-bit representations (Nagel et al., 2021). With the thriving of LLMs, many efforts (Tseng et al., 2024; Lin et al., 2024b) have focused on adapting quantization techniques for LLMs. For example, GPTQ (Frantar et al., 2022) improves upon the classic OBQ (Frantar & Alistarh, 2022) by enabling efficient post-training quantization on large-scale parameters and can outperform the common RTN baseline. Moreover, LLMs also contain outliers, where a small number

of elements exhibit disproportionately large magnitudes and heavily influence downstream performance. To address this, LLM.int8() (Dettmers et al., 2022) introduces a mixed-precision scheme that preserves outliers in higher precision. Later, AWQ (Lin et al., 2024a) proposes to employ smoothing factors to transfer outliers from weights to activations, thus allowing for 8-bit weight quantization. SmoothQuant (Xiao et al., 2023) further trades off smoothing between weights and activations to achieve W8A8 quantization. To push toward even lower bit-widths, recent works (Chee et al., 2023; Hu et al., 2025; Sun et al., 2024) have predominantly leveraged the Hadamard transformation to flatten the weight distributions before quantization. For instance, QuaRot (Ashkboos et al., 2024) applies random rotation as a preprocessing step, enabling quantization even to W4A4KV4 while maintaining performance. SpinQuant (Liu et al., 2024) and FlatQuant (Sun et al., 2024) further extend this idea by introducing learnable rotation matrices to further enhance quantization performance.

**Network Pruning for LLMs.** Network pruning reduces computational and memory costs by eliminating weights that contribute little to the final prediction (LeCun et al., 1989; Han et al., 2015; Frankle & Carbin, 2018; Zhang et al., 2024). Early pruning methods primarily relied on magnitude-based criteria, which proved effective for small-scale networks. However, these simple approaches often struggle to maintain accuracy when applied to LLMs. To address this, a variety of methods have been developed to either refine the pruning process or introduce more advanced selection criteria. For instance, LLM-Pruner (Ma et al., 2023) proposes to remove coupled components followed by LoRA (Hu et al., 2022) finetuning to restore accuracy. SparseGPT (Frantar & Alistarh, 2023) introduces a one-shot pruning method based on OBD (LeCun et al., 1989), enabling efficient pruning without additional retraining. WANDA (Sun et al., 2023) demonstrates that information contained in activations is crucial for LLMs pruning, and introduces a simple yet effective scoring metric for activation-aware sparsity.

**Joint Quantization and Sparsification.** Before the rise of LLMs, several early works explored joint quantization and pruning on small networks. For instance, DJPQ (Wang et al., 2020) solves an optimization problem via gradient descent to balance sparsity and quantization error. OBQ (Frantar & Alistarh, 2022) proposes a unified framework that simultaneously considers both pruning and quantization. In the context of LLMs, JSQ (Guo et al., 2024) adopts simulated annealing to identify optimal activation editing policies, and can achieve W8A8 quantization with 50% sparsity. Moreover, one recent work (Harma et al., 2024) also provides a theoretical analysis suggesting that pruning followed by quantization is the optimal compression order. Despite these advancements, existing techniques still fall short in achieving aggressive compression levels such as W4A4KV4 with 50% sparsity, leaving room for further improvement in this domain.

## 3 MOTIVATION

As shown in Fig. 1(a), relying on a single method such as quantization or pruning is rapidly approaching its compression limits. For instance, solely decreasing the quantization bit-width or increasing the pruning ratio leads to drastic performance degradation. In contrast, since different compression techniques are largely orthogonal in nature (Guo et al., 2024), combining them effectively presents a potential direction to "squeeze out" additional efficiency. For instance, as shown in Appendix B, the W4A4KV4 quantized Llama2-7B model in QuaRot (Ashkboos et al., 2024) naturally exhibits 14.28% average layer sparsity. Moreover, recent hardware advances have already supported INT4 sparse GEMM, which can achieve faster execution than dense INT4 kernels in practice. These observations motivate us to explore how to jointly leverage quantization and sparsity for more aggressive and practical LLM compression.

However, realizing an effective joint quantization and sparsification scheme is notoriously challenging due to their inherently conflicting nature. Specifically, quantization typically favors a compact numerical range of weights to minimize quantization error. For example, recent 4-bit quantization methods (Ashkboos et al., 2024; Liu et al., 2024; Sun et al., 2024) commonly adopt Hadamard transformation to rotate weights into smoother distributions for suppressing outliers before quantization. While such rotation is beneficial for quantization, it is detrimental to sparsity, which instead prefers weight distributions that exhibit large numerical disparities to better encourage sparsity. As demonstrated in Sec. 5.1, naively applying sparsification on top of rotated weights leads to severe performance degradation.

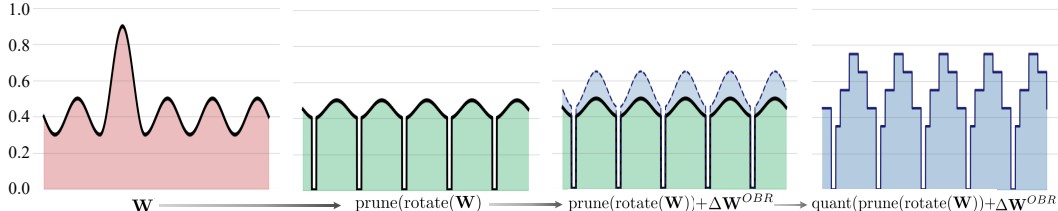

Figure 2: Given original LLM weights $\mathbf{W}$, we first apply a rotation to smooth out outliers, followed by pruning to introduce sparsity. The proposed OBR is employed to compute optimal compensation, which is added to the unpruned elements to mitigate the conflict between pruning and quantization. Finally, quantization is applied to obtain the sparse and quantized LLM weights.

## 4 OPTIMAL BRAIN RESTORATION

In this work, we propose the Optimal Brain Restoration (OBR) framework, which adjusts weight distributions to reconcile the conflicting demands of pruning and quantization. Following previous practices (Harma et al., 2024; Guo et al., 2024), we adopt an order of pruning-then-quantization. As shown in Fig. 2, the overall process to generate low-bit and sparse weights using the proposed OBR can be formalized as:

$$\hat{\mathbf{W}} = \text{quant}(\text{prune}(\text{rotate}(\mathbf{W})) + \Delta\mathbf{W}^{OBR}), \tag{1}$$

where $\mathbf{W}$ is the original LLM weights, $\Delta\mathbf{W}^{OBR}$ is the compensation derived from OBR. In the following, we start in Sec. 4.1 by defining the necessary notations and objective function. Then we detail the generic formulation of the proposed OBR in Sec. 4.2, followed by the specific instantiations for quantization and pruning in Sec. 4.3.

### 4.1 OBJECTIVE APPROXIMATION

Given the weight matrix $\mathbf{W} \in \mathbb{R}^{C_{out} \times C_{in}}$ in one standard linear layer and $\mathbf{X} \in \mathbb{R}^{C_{in} \times L}$ being the input activation representing the dataset's statistics, our work employs the following classic optimization objective (LeCun et al., 1989; Frantar & Alistarh, 2022) which minimizes the perturbation of downstream task loss:

$$\min \quad \mathbb{E}[\Delta\mathcal{L}] = \mathbb{E}[\mathcal{L}(\mathbf{X}, \mathbf{W} + \Delta\mathbf{W}) - \mathcal{L}(\mathbf{X}, \mathbf{W})], \tag{2}$$

where $\Delta\mathbf{W}$ is the perturbation on $\mathbf{W}$, $\mathcal{L}$ is the downstream task loss.

To solve the optimization problem in Eq. (2), we first simplify the objective function. In detail, applying Taylor series on $\mathcal{L}(\mathbf{X}, \mathbf{W} + \Delta\mathbf{W})$ at $\mathbf{W}$ drives:

$$\Delta\mathcal{L} = \langle \nabla_{\mathbf{W}}\mathcal{L}(\mathbf{X}, \mathbf{W}), \Delta\mathbf{W} \rangle + \frac{1}{2}\text{vec}(\Delta\mathbf{W})\mathbf{H}_{\text{full}}\text{vec}(\Delta\mathbf{W})^{\top} + \mathcal{O}(\|\Delta\mathbf{W}\|^3), \tag{3}$$

where $\nabla_{\mathbf{W}}\mathcal{L}(\mathbf{X}, \mathbf{W})$ is the gradient, $\text{vec}(\cdot) : \mathbb{R}^{C_{out} \times C_{in}} \rightarrow \mathbb{R}^{1 \times C_{out}C_{in}}$ is the vectorisation operator, and $\mathbf{H}_{\text{full}} \triangleq \frac{\partial^2 \mathcal{L}}{\partial\text{vec}(\mathbf{W})\partial\text{vec}(\mathbf{W})^{\top}} \in \mathbb{R}^{C_{out}C_{in} \times C_{out}C_{in}}$ is the layer-wise Hessian.

Assume that the model has been fully trained and reaches a local minima, so the $\nabla_{\mathbf{W}}\mathcal{L}(\mathbf{X}, \mathbf{W}) \approx 0$. Further ignoring the last high order terms, Eq. (3) can be approximated into:

$$\Delta\mathcal{L} \approx \frac{1}{2}\text{vec}(\Delta\mathbf{W})\mathbf{H}_{\text{full}}\text{vec}(\Delta\mathbf{W})^{\top}. \tag{4}$$

Despite the above preliminary approximation, computing $\mathbf{H}_{\text{full}}$ exactly is still infeasible in LLMs due to the $\mathcal{O}((C_{out}C_{in})^2)$ complexity, we thus following previous works (Frantar & Alistarh, 2022) and estimate $\mathbf{H}_{\text{full}}$ as:

$$\mathbf{H}_{\text{full}} \approx \mathbf{G} \otimes \mathbf{H}, \tag{5}$$

where $\mathbf{G} \in \mathbb{R}^{C_{out} \times C_{out}}$ is the output-side curvature matrix which depicts the second-order sensitivity among output channels, $\mathbf{H} \triangleq 2\mathbf{X}\mathbf{X}^{\top} \in \mathbb{R}^{C_{in} \times C_{in}}$ is the empirical Fisher matrix, and $\otimes$ denotes the Kronecker product.

Based on Eq. (5), we propose to decouple the row-wise correlation of output channels in $\mathbf{H}_{\text{full}}$ by approximating $\mathbf{G}$ as an Identity matrix $\mathbf{I}$ to make $\mathbf{H}_{\text{full}} \approx \mathbf{I} \otimes \mathbf{H}$ completely tractable. Finally, the original objective can be simplified into the following $C_{out}$ independent optimization sub-problems:

$$\min \quad \mathbb{E}[\frac{1}{2}\text{vec}(\Delta\mathbf{W})(\mathbf{I} \otimes \mathbf{H})\text{vec}(\Delta\mathbf{W})^\top] = \frac{1}{2}\sum_{i=1}^{C_{out}} \mathbb{E}[\Delta\mathbf{w}_i\mathbf{H}\Delta\mathbf{w}_i^\top], \tag{6}$$

where $\Delta\mathbf{w}_i \in \mathbb{R}^{1 \times C_{in}}$ is the $i$-th row of $\Delta\mathbf{W}$. Intuitively, Eq. (6) quantifies the impact of weight changes on the final downstream performance. For example, when $\mathbf{H}$ is large, even a small change in weights can result in large differences for downstream tasks.

## 4.2 SOLUTION AND FRAMEWORK

To solve the simplified objective in Eq. (6), our proposed OBR employs the Group Error Compensation to optimally adjust weight distributions by shifting information from error-sensitive groups to the other robust ones. Since the rotation matrix acts on both $\mathbf{W}$ and $\mathbf{X}$, and thus cancels out during multiplication, in the following sections, we will omit the rotation operation and directly denote $\mathbf{W}$ as the rotated matrix for notational clarity.

Let $\mathcal{J}_i = \frac{1}{2}\Delta\mathbf{w}_i\mathbf{H}\Delta\mathbf{w}_i^\top$ denote the $i$-th sub-problem, we begin by partitioning the elements of the $i$-th row $\Delta\mathbf{w}_i$ into two disjoint groups using two index sets, *i.e.*, the retain set $R_i$ and the eviction set $E_i$, where $R_i \cup E_i = \{1, \ldots, C_{in}\}$ and $R_i \cap E_i = \emptyset$. The retain set $R_i$ collects weights that are less affected by compression, *e.g.*, unpruned or less quantization-distorted, whereas the eviction set $E_i$ corresponds to the indices of elements that are susceptible to compression effects. For clarity, we will omit the row index $i$ in the following.

With this grouping, our key idea is to compensate for compression-induced errors $\mathbf{e}_E$ in eviction set $E$ by transferring its lost information to a more robust retain set $R$. To enable this, we reorder the perturbation vector $\Delta\mathbf{w}$ into $[\Delta\mathbf{w}_R, \Delta\mathbf{w}_E]$. Then the sub-problem becomes:

$$\underset{\Delta\mathbf{w}_R}{\arg\min} \quad \mathcal{J} = \frac{1}{2}\Delta\mathbf{w}\mathbf{H}\Delta\mathbf{w}^\top = \frac{1}{2}[\Delta\mathbf{w}_R \quad \mathbf{e}_E] \begin{bmatrix} \mathbf{H}_{RR} & \mathbf{H}_{RE} \\ \mathbf{H}_{ER} & \mathbf{H}_{EE} \end{bmatrix} \begin{bmatrix} \Delta\mathbf{w}_R^\top \\ \mathbf{e}_E^\top \end{bmatrix}. \tag{7}$$

Since Eq. (7) is an unconstrained optimization problem, we can directly obtain the closed-form solution by taking the partial derivatives *w.r.t.* $\Delta\mathbf{w}_R$, *i.e.*, $\nabla_{\Delta\mathbf{w}_R}\mathcal{J} = \mathbf{H}_{RR}\Delta\mathbf{w}_R + \mathbf{H}_{RE}\mathbf{e}_E \triangleq 0$. Then the optimal solution for $\Delta\mathbf{w}_R$ which minimizes the row-wise error can be derived as:

$$\Delta\mathbf{w}_R^\star = -\mathbf{H}_{RR}^{-1}\mathbf{H}_{RE}\mathbf{e}_E. \tag{8}$$

In Fig. 3(a), we give an example on how to extract sub-Hessian $\mathbf{H}_{RR}$ and $\mathbf{H}_{RE}$ from $\mathbf{H}$. According to the above formulation, the error in set $E$ is theoretically zero guaranteed by the closed-form solution. Since the retain set $R$ is assumed to be robust against compression-related errors, the total error can be decreased through transferring information from $E$ to $R$. Notably, Eq. (8) also offers a strong explanation that the Hessian actually serves as a "bridge" for error propagation between different groups. Specifically, in Eq. (8), the $\mathbf{e}_E$ is first projected from $E$'s space to the shared space via $\mathbf{H}_{RE}$, followed by the mapping to the $R$'s space through $\mathbf{H}_{RR}^{-1}$, and the negative sign denoting the correction direction.

## 4.3 SPECIFIC IMPLEMENTATION

In this section, we apply the generic closed-form solution in Eq. (8) to the specific implementation for sparsification and quantization.

**OBR for Sparsification.** As shown in Fig. 3(b), given the 0-1 mask from existing pruning algorithms, we define retain set $R_1$ as the unpruned slots, and eviction set $E_1$ as the pruned ones. In this way, the information loss due to pruning on set $E_1$ can be compensated by transferring to set $R_1$. Formally, since the pruning error on set $E_1$ is $\mathbf{e}_{E_1}^{prune} = \mathbf{w}_{E_1}$, using Eq. (8), the optimal OBR compensation for pruning can be derived as:

$$\Delta\mathbf{w}_{R_1}^{prune} = -\mathbf{H}_{R_1 R_1}^{-1}\mathbf{H}_{R_1 E_1}\mathbf{w}_{E_1}. \tag{9}$$

We then add $\Delta\mathbf{w}_{R_1}^{prune}$ to the unpruned elements $\mathbf{w}_{R_1}$ to obtain the OBR-compensated sparse weight $\bar{\mathbf{w}} = [\mathbf{w}_{R_1} + \Delta\mathbf{w}_{R_1}^{prune}, \mathbf{0}]$. After that, we perform another round of OBR on $\bar{\mathbf{w}}$ to further consider the incoming quantization error. Details are given below.

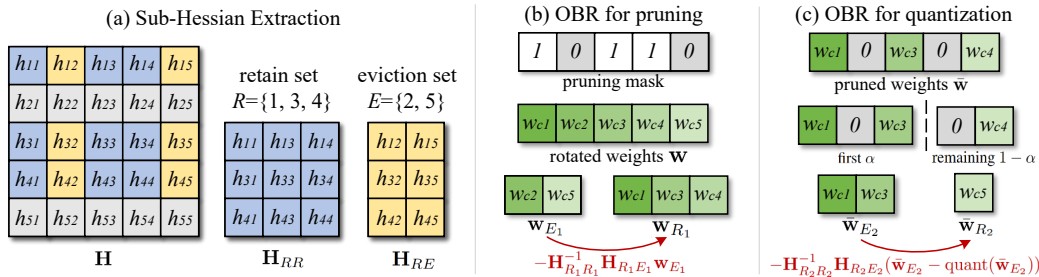

Figure 3: (a) Given a Hessian approximation $\mathbf{H}$, we extract the submatrices $\mathbf{H}_{RR}$ and $\mathbf{H}_{RE}$ based on the index sets $R$ and $E$. (b) The rotated dense weights are partitioned into $R_1$ and $E_1$ according to the binary pruning mask, followed by OBR to transfer information from $\mathbf{w}_{E_1}$ to $\mathbf{w}_{R_1}$. (c) The unpruned index set $R_1$ is further divided into two groups: the first $\alpha$ fraction assigned to set $E_2$, the remaining $1 - \alpha$ to set $R_2$. OBR is used to compensate for quantization error in $E_2$.

**OBR for Quantization.** Different from pruning where the retain set and eviction set can be naturally obtained from the pruning mask, in quantization, we need to manually assign the grouping to obtain $R_2$ and $E_2$ for compensation with OBR. Thanks to the flat distribution introduced by Hadamard rotation, we find the discrepancy among unpruned elements is actually small (see Fig. 6). Inspired by this observation, we propose to take the first $\alpha$ proportion of elements in set $R_1$ as the set $E_2$, and the remaining $1 - \alpha$ proportion of elements as the set $R_2$. In other words, $|R_2| + |E_2| = |R_1|$, where $|\cdot|$ is the number of elements. In Fig. 3(c), given quantization error on set $E_2$ as $\mathbf{e}_{E_2}^{quant} = \bar{\mathbf{w}}_{E_2} - \text{quant}(\bar{\mathbf{w}}_{E_2})$, we can obtain the OBR compensation for quantization as follows:

$$\Delta \mathbf{w}_{R_2}^{quant} = -\mathbf{H}_{R_2 R_2}^{-1} \mathbf{H}_{R_2 E_2}(\bar{\mathbf{w}}_{E_2} - \text{quant}(\bar{\mathbf{w}}_{E_2})). \tag{10}$$

Considering both quantization and pruning, the overall OBR-processed weights can be formalized as:

$$\hat{\mathbf{w}} = \text{quant}([\mathbf{w}_{R_2} + \Delta \mathbf{w}_{R_2}^{prune} + \Delta \mathbf{w}_{R_2}^{quant}, \quad \mathbf{w}_{E_2} + \Delta \mathbf{w}_{E_2}^{prune}, \quad \mathbf{0}]), \tag{11}$$

where $\Delta \mathbf{w}_{R_2}^{prune}$ and $\Delta \mathbf{w}_{E_2}^{prune}$ denote indexing from $\Delta \mathbf{w}_{R_1}^{prune}$ using $R_2$ and $E_2$, and $\hat{\mathbf{w}}$ is the final joint low-bit and sparse weights. Algo. 1 provides more details of our proposed OBR.

**CUDA Kernel Implementation.** After transforming LLMs to both sparse and low-bit using the proposed OBR, we implement corresponding GEMM with the CUTLASS library[1]. To align with hardware supported quantization and sparsity, we perform 2:4 semi-structured sparsity and INT4 quantization on the weights $\mathbf{W}$, and use INT4 quantization for activations $\mathbf{X}$ and KV caches.

## 5 EXPERIMENTS

**Datasets and Models.** We evaluate the proposed OBR framework on various open-source LLM families, including Llama2 (7B/13B/70B) (Touvron et al., 2023), Llama3 (8B/70B) (Dubey et al., 2024), and Qwen2.5-Instruct(7B/32B) (QwenTeam, 2024). To comprehensively assess the effectiveness of our method, we conduct experiments on both zero-shot classification and language modeling tasks. For zero-shot evaluation, we report accuracy on commonly used benchmarks including PIQA (Bisk et al., 2020), BoolQ (Clark et al., 2019), HellaSwag (Zellers et al., 2019), ARC-easy (Clark et al., 2018), ARC-challenge (Clark et al., 2018), and WinoGrande (Sakaguchi et al., 2021). In addition, we also follow prior LLM compression works (Sun et al., 2023) and evaluate the perplexity on the WikiText2 test set (Merity et al., 2016).

**Baselines.** We compare our method against a range of competitive baselines under sub-4-bit compression settings. Specifically, the full-precision model is included as an upper bound for reference. We also evaluate against quantization-only baselines (Ashkboos et al., 2024; Liu et al., 2024) under equivalent bit-widths, e.g., a W4A4 model with 50% sparsity is compared to a W3A4 quantized model. In addition, we include a simple baseline that directly combines existing quantization and pruning techniques without any specially designed compensation. Furthermore, following the extension described in (Frantar & Alistarh, 2023), we adopt SparseGPT combined with GPTQ as a strong joint sparsity-quantization baseline for comparison.

[1] https://github.com/NVIDIA/cutlass

Table 1: Comparison of perplexity score on WikiText2 and accuracy on zero-shot common sense reasoning tasks with Llama2(7B/13B/70B) and Llama3(8B/70B) families. †Since the Llama3-70B is sensitive to quantization as demonstrated in (Ashkboos et al., 2024), we keep the KV cache being 16-bit for acceptable performance. The best and the second best results are in red and blue.

| Model | Method | #Bits W-A-KV | Sparsity ratio | PIQA (↑) | BoolQ (↑) | HellaS. (↑) | Arc-e (↑) | Arc-c (↑) | WinoG. (↑) | Avg. (↑) | Wiki2 (↓) |
|---|---|---|---|---|---|---|---|---|---|---|---|
| 2-7B | Floating-point | 16-16-16 | 0% | 79.11 | 77.71 | 76.02 | 74.49 | 46.33 | 69.14 | 70.47 | 5.47 |
| | QuaRot(quant-only) | 3-4-4 | 0% | 51.96 | 39.72 | 29.25 | 31.36 | 23.46 | 52.33 | 38.01 | 132.97 |
| | QuaRot+WANDA | 4-4-4 | 50% | 50.27 | 37.83 | 25.81 | 25.00 | 27.73 | 49.25 | 35.98 | 5868.24 |
| | SparseGPT+GPTQ | 4-4-4 | 50% | 63.38 | 63.27 | 47.71 | 50.93 | 29.44 | 54.70 | 51.57 | 12.94 |
| | OBR_RTN | 4-4-4 | 50% | 68.77 | 66.39 | 55.46 | 55.98 | 32.17 | 60.22 | 56.49 | 9.23 |
| | OBR_GPTQ | 4-4-4 | 50% | 68.93 | 67.31 | 58.22 | 55.93 | 34.22 | 61.48 | 53.45 | 8.40 |
| 2-13B | Floating-point | 16-16-16 | 0% | 80.52 | 80.55 | 79.37 | 77.48 | 49.15 | 72.14 | 73.20 | 4.88 |
| | QuaRot(quant-only) | 3-4-4 | 0% | 55.01 | 62.26 | 30.00 | 31.10 | 22.44 | 51.07 | 41.98 | 72.53 |
| | QuaRot+WANDA | 4-4-4 | 50% | 51.36 | 38.29 | 26.40 | 26.18 | 27.56 | 49.49 | 36.54 | 2289.41 |
| | SparseGPT+GPTQ | 4-4-4 | 50% | 71.27 | 70.83 | 60.99 | 61.87 | 36.60 | 62.90 | 60.74 | 7.89 |
| | OBR_RTN | 4-4-4 | 50% | 72.74 | 69.17 | 63.85 | 65.95 | 38.31 | 64.17 | 62.37 | 7.29 |
| | OBR_GPTQ | 4-4-4 | 50% | 72.91 | 71.25 | 64.74 | 65.57 | 37.88 | 63.22 | 62.60 | 7.06 |
| 2-70B | Floating-point | 16-16-16 | 0% | 82.70 | 83.76 | 83.81 | 81.06 | 57.25 | 77.98 | 77.76 | 3.32 |
| | QuaRot(quant-only) | 3-4-4 | 0% | 67.74 | 66.27 | 56.55 | 50.67 | 30.63 | 62.43 | 55.72 | 8.19 |
| | QuaRot+WANDA | 4-4-4 | 50% | 51.52 | 38.56 | 27.67 | 27.06 | 23.21 | 50.04 | 36.34 | 169.67 |
| | SparseGPT+GPTQ | 4-4-4 | 50% | 79.11 | 76.79 | 77.20 | 77.61 | 51.19 | 73.95 | 72.64 | 4.78 |
| | OBR_RTN | 4-4-4 | 50% | 78.67 | 75.93 | 76.09 | 77.57 | 51.96 | 74.51 | 72.45 | 4.84 |
| | OBR_GPTQ | 4-4-4 | 50% | 79.22 | 76.91 | 77.23 | 77.53 | 50.68 | 74.11 | 72.61 | 4.69 |
| 3-8B | Floating-point | 16-16-16 | 0% | 80.85 | 80.98 | 79.17 | 77.74 | 53.24 | 73.40 | 74.23 | 6.13 |
| | QuaRot(quant-only) | 3-4-4 | 0% | 55.28 | 39.72 | 30.78 | 30.72 | 21.76 | 50.36 | 38.10 | 196.23 |
| | QuaRot+WANDA | 4-4-4 | 50% | 49.62 | 37.95 | 26.42 | 27.02 | 23.98 | 47.83 | 35.47 | 1927.29 |
| | SparseGPT+GPTQ | 4-4-4 | 50% | 66.21 | 65.41 | 53.58 | 50.67 | 29.52 | 57.22 | 53.77 | 16.40 |
| | OBR_RTN | 4-4-4 | 50% | 67.95 | 64.98 | 54.06 | 52.57 | 30.89 | 55.96 | 54.40 | 14.47 |
| | OBR_GPTQ | 4-4-4 | 50% | 66.87 | 65.23 | 55.41 | 54.63 | 30.03 | 58.80 | 55.16 | 13.92 |
| 3-70B† | Floating-point | 16-16-16 | 0% | 84.49 | 85.38 | 84.96 | 86.11 | 64.16 | 80.51 | 80.93 | 2.85 |
| | QuaRot(quant-only) | 3-4-16 | 0% | 52.77 | 51.99 | 30.65 | 31.23 | 23.12 | 50.51 | 40.05 | 80.25 |
| | QuaRot+WANDA | 4-4-16 | 50% | 50.84 | 37.83 | 26.25 | 25.38 | 26.96 | 45.70 | 35.49 | 23245.17 |
| | SparseGPT+GPTQ | 4-4-16 | 50% | 60.12 | 52.81 | 35.02 | 38.30 | 23.29 | 53.51 | 43.84 | 41.39 |
| | OBR_RTN | 4-4-16 | 50% | 61.92 | 56.54 | 37.81 | 43.77 | 25.17 | 52.01 | 46.20 | 33.38 |
| | OBR_GPTQ | 4-4-16 | 50% | 67.36 | 64.40 | 55.26 | 55.64 | 33.11 | 50.59 | 55.96 | 16.69 |

**Implementation Details.** Since our OBR framework, as well as most other pruning and quantization methods (Frantar et al., 2022; Frantar & Alistarh, 2023; Sun et al., 2023), require calibration data to estimate input statistics, we follow standard practice and use 128 samples from WikiText2 with a sequence length of 2048 as the calibration set. For the Hadamard transformation, we test our OBR on rotation matrices from various existing works, including QuaRot (Ashkboos et al., 2024), SpinQuant (Liu et al., 2024), and FlatQuant (Sun et al., 2024). In addition, as our OBR treats pruning mask and quantizer as givens, it is potentially compatible with different pruning and quantization methods. Therefore, for pruning, we adopt the 0-1 mask generated by WANDA (Sun et al., 2023) as the default setting due to its strong performance and training-free nature. We will further discuss OBR's generality across other pruning algorithms in Sec. 5.2. For the grouping ratio $\alpha$ in OBR quantization, we simply use $\alpha = 50\%$ as the default setting for all setups. For quantization, we include both the simple Round-To-Nearest (RTN) quantizer to obtain OBR_RTN, and the more advanced GPTQ (Achiam et al., 2023) quantizer for OBR_GPTQ as an extension.

## 5.1 EXPERIMENT RESULTS

**Main Results.** As shown in Tab. 1, the QuaRot (quant-only), which relies solely on quantization for compression, suffers from severe performance degradation under 4-bit, *e.g.*, 132.97 perplexity for W3A4KV4 quantized Llama2-7B model. Furthermore, effectively combining quantization and sparsity is non-trivial. For example, directly combining the existing quantization method Quarot (Ashkboos et al., 2024) with the pruning method WANDA (Sun et al., 2023) leads to unacceptable performance. For joint quantization and sparsification comparison, our OBR with a simple RTN quantizer can achieve even better performance than SparseGPT+GPTQ in most cases. For example, our OBR_RTN achieves even 3.71 better perplexity compared to SparseGPT+GPTQ on the Llama2-7B model. When using the more advanced quantizer GPTQ, our OBR_GPTQ can achieve a further 0.83 perplexity improvement. These experimental results demonstrate the effectiveness of the proposed OBR framework across different LLMs and tasks.

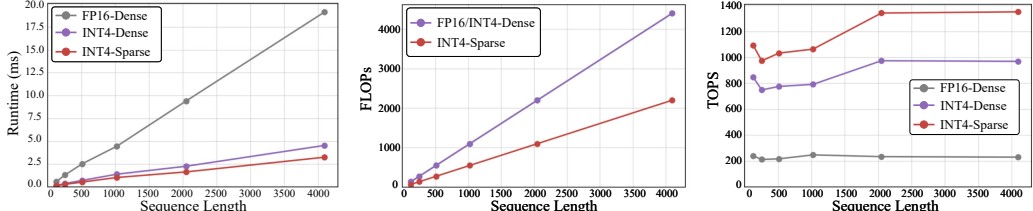

Figure 4: Comparison on runtime, FLOPs, and TOPS across different sequence lengths. We evaluate the performance of FP16-Dense, INT4-Dense, and INT4 2:4 Sparse GEMM on a single NVIDIA A100-SXM4-80GB GPU. The GEMM computation follows a typical LLM inference setting, where the weight matrix is $\mathbf{W} \in \mathbb{R}^{4096 \times 4096}$ and the input activation is $\mathbf{X} \in \mathbb{R}^{32 \times seq\_len \times 4096}$.

Table 2: Comparison under other quantization bit-widths on WikiText2 perplexity (wiki2) and average zero-shot accuracy (0-shot) using the Llama2-7B model.

| Method | sparisty | W4A8KV8 | | W4A16KV16 | |
|---|---|---|---|---|---|
| | | wiki2↓ | 0-shot↑ | wiki2↓ | 0-shot↑ |
| Quarot(quant-only) | 0% | 80.525 | 39.98 | 80.25 | 40.04 |
| Quarot+WANDA | 50% | 5278.13 | 35.95 | 5272.07 | 35.92 |
| SparseGPT+GPTQ | 50% | 8.53 | 59.41 | 8.53 | 59.47 |
| OBR_RTN | 50% | 7.24 | 62.16 | 7.24 | 62.27 |
| OBR_GPTQ | 50% | 6.87 | 63.39 | 6.86 | 63.33 |

**Practical Speedups.** Given that recent GPU architectures such as Ampere and Hopper have naively supported INT4-sparse GEMM kernels, we compare the efficiency on batched matrix multiplication with other two baselines, *i.e.*, INT4-dense and FP16-dense GEMM, in terms of latency, FLOPs, and TOPS. In Fig. 4, as input token length increases, the latency advantage of INT4+2:4 sparse GEMM becomes more pronounced. For example, at a sequence length of 4096, the INT4+2:4 sparse GEMM achieves a 5.9× speedup over FP16-dense and a 1.4× speedup over INT4-dense GEMM. Furthermore, thanks to the 50% sparsity, INT4+2:4 sparse GEMM reduces theoretical FLOPs by 2× compared to its dense counterpart. Finally, when the GPU compute resources are fully saturated, *i.e.*, with sequence length> 2048, the INT4+2:4 GEMM also achieves higher throughput in terms of TOPS. These results highlight the efficiency potential of low-bit sparse GEMM in real-world deployment compared to classic dense low-bit matrix multiplication.

**Comparison on other Bits.** We further evaluate the OBR framework under more bit-width configurations. Given that LLMs are known to be memory-bound, we keep the weights quantized to low precision, *i.e.*, 4-bit, while varying the activation and KV cache bit-width. Tab. 2 presents the results for W4A8KV8 and W4A16KV16 (weight-only quantization) settings. One can see that our OBR consistently outperforms all competitive baselines. Notably, OBR_RTN with W4A8KV8+50% sparsity even surpasses weight-only quantization of SparseGPT+GPTQ by 1.29 perplexity. These results demonstrate the generality and effectiveness of OBR across different quantization bit-widths.

**Results with SpinQuant.** To further validate the generality of other rotation schemes, we apply OBR to SpinQuant (Liu et al., 2024), which introduces learnable rotation matrices for improved performance. Similar to the setup of QuaRot, we treat the rotation matrix as given and do not learn a dedicated rotation matrix for the joint quantization-sparsification setting. As shown in Tab. 3, our method achieves notable improvements over other competitive baselines *e.g.*, OBR_RTN achieves 7.69% average accuracy improvement against SparseGPT+GPTQ on zero-shot evaluation with Llama2-7B. Since the quantization-only W3A4KV4 baseline employs the rotation matrices specifically trained for quantization, our method is slightly inferior due to the task gap. We believe learning rotation matrices specifically for low-bit and sparse setups holds potential for further improvement.

**Other Sparsity Patterns.** Semi-structured pruning, such as 2:4 sparsity, is now well-supported by modern hardware to achieve practical acceleration. To this end, we further include comparisons under semi-structured pruning settings in Tab. 4. One can see that the advantages of our OBR become more apparent as the compression becomes more challenging. In detail, both OBR_RTN and OBR_GPTQ consistently outperform other baselines under given setups. For example, under the challenging W4A4KV4+2:4 sparse setup, our OBR_RTN reduces perplexity by 18.8 and improves the average accuracy on zero-shot evaluation by 5.86% over the SparseGPT+GPTQ. These promising results demonstrate the effectiveness of OBR in joint low-bit quantization and semi-structured sparsity.

Table 3: Comparison of perplexity on WikiText2 and average accuracy on 0-shot commonsense reasoning tasks under SpinQuant (Liu et al., 2024) rotated weights.

| Method | bits | sparsity | Llama2-7B | | Llama2-13B | | Llama2-70B | | Llama3-8B | | Llama3-70B | |
|---|---|---|---|---|---|---|---|---|---|---|---|---|
| | | | wiki2↓ | 0-shot↑ | wiki2↓ | 0-shot↑ | wiki2↓ | 0-shot↑ | wiki2↓ | 0-shot↑ | wiki2↓ | 0-shot↑ |
| SpinQuant(quant-only) | 3-4-4 | 0% | 8.24 | 58.95 | 6.39 | 66.78 | 4.21 | 74.09 | 10.50 | 60.29 | 9.64 | 63.64 |
| SpinQuant+WANDA | 4-4-4 | 50% | 1589.54 | 36.17 | 648.59 | 35.94 | 26.99 | 43.77 | 703.05 | 39.05 | 18565.64 | 36.27 |
| SparseGPT+GPTQ | 4-4-4 | 50% | 22.57 | 45.42 | 8.47 | 57.39 | 4.75 | 72.75 | 16.37 | 53.67 | 21.74 | 51.14 |
| OBR_RTN | 4-4-4 | 50% | 10.40 | 53.11 | 7.57 | 60.72 | 4.71 | 72.85 | 13.10 | 55.22 | 18.18 | 49.30 |
| OBR_GPTQ | 4-4-4 | 50% | 10.70 | 53.45 | 7.17 | 61.50 | 4.60 | 72.88 | 13.34 | 55.28 | 11.60 | 60.64 |

Table 4: Comparison on 4:8 and 2:4 sparsity with Llama2-7B models. The included baselines have all been quantized using QuaRot W4A4KV4 configuration.

| Method | sparsity | wiki2↓ | 0-shot↑ |
|---|---|---|---|
| Floating-point | - | 5.47 | 70.46 |
| SparseGPT+GPTQ | 4:8 | 20.29 | 44.99 |
| OBR_RTN | 4:8 | 11.45 | 51.60 |
| OBR_GPTQ | 4:8 | 10.61 | 52.02 |
| SparseGPT+GPTQ | 2:4 | 34.76 | 40.52 |
| OBR_RTN | 2:4 | 15.96 | 46.38 |
| OBR_GPTQ | 2:4 | 13.32 | 48.67 |

Figure 5: Applying the proposed OBR to WANDA (Sun et al., 2023) pruning algorithm in single compression tasks.

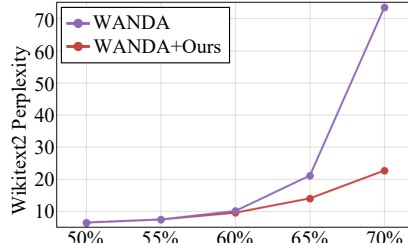

Table 5: Ablation on different pruning masks under W4A4KV4+50% sparsity using Llama2-7B and QuaRot rotation.

| pruning metirc | wiki2↓ | 0-shot↑ |
|---|---|---|
| Magnitude: $|\mathbf{W}|$ | 8.92 | 56.51 |
| SparseGPT: $[|\mathbf{W}|^2/diag(\mathbf{H}^{-1})]$ | 9.28 | 55.45 |
| WANDA: $|\mathbf{W}| \cdot |\mathbf{X}|$ | 8.40 | 53.45 |

Table 6: Ablation on partition ratio $\alpha$.

| $\alpha$ | $1-\alpha$ | Llama2-7B | | Llama2-13B | |
|---|---|---|---|---|---|
| | | wiki2↓ | 0-shot↑ | wiki2↓ | 0-shot↑ |
| 75% | 25% | 9.96 | 53.56 | 7.70 | 60.22 |
| 50% | 50% | 9.23 | 56.49 | 7.29 | 62.37 |
| 25% | 75% | 9.07 | 57.06 | 7.09 | 63.20 |
| 20% | 80% | 8.89 | 56.79 | 7.43 | 61.53 |

## 5.2 ABLATION STUDIES

**Different Pruning Masks.** In the proposed OBR framework, the pruning mask is treated as a given, making our method compatible with various existing pruning algorithms. In the above main experiments, we primarily adopt masks generated from WANDA (Sun et al., 2023) pruning. To further evaluate the effectiveness of other pruning metrics, we report in Tab. 5 the results using magnitude-based, SparseGPT-based (Frantar & Alistarh, 2023), and even Random pruning masks. Thanks to the error compensation from OBR, even the naive magnitude metric can achieve satisfactory performance. These results demonstrate the robustness of the proposed method across different pruning metrics.

**Partition Ratios for OBR Quantization.** For quantization error compensation in OBR, we adopt a simple strategy that splits weights into two groups with the first $\alpha$ proportion as the eviction set $E_2$ and the remaining as the retain set $R_2$, followed by the OBR error transfer. To further understand how the partitioning ratio affects error compensation, we conduct an ablation study with different $\alpha$. As shown in Tab. 6, transferring the error from 20% elements to the remaining 80% leads to a performance drop due to an insufficient compensating number. Conversely, migrating 75% of the error to only 25% of the elements also yields suboptimal results due to low-quality compensation. As a trade-off, we adopt a 50% partitioning ratio for constructing $E_2$ and $R_2$ as our final design.

## 5.3 DISCUSSION

**OBR for Pruning Only.** As shown in Sec. 4.3, the proposed OBR can be potentially applied to a single compression task to compensate for errors produced by a given compression algorithm. To this end, we first extend our OBR framework to the pruning-only task. Specifically, we apply the proposed OBR to WANDA (Sun et al., 2023) by compensating for post-pruning weight distortions. The perplexity results on WikiText2 under different sparsity ratios are reported in Fig. 5. Equipped with our OBR, WANDA consistently achieves lower perplexity under given sparsity levels. For instance, at 60% sparsity, WANDA+OBR improves perplexity by 0.53 compared to the original WANDA, and this performance gain becomes more pronounced when sparsity increases. These

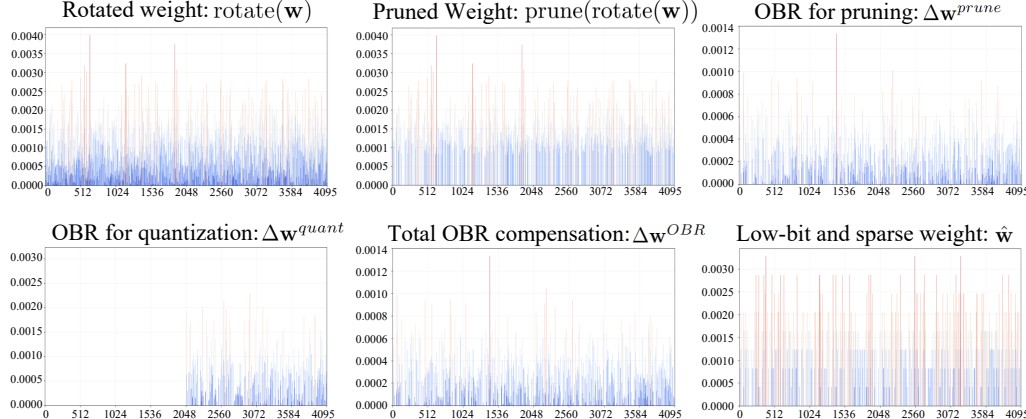

Figure 6: Distribution visualization of different stages in the OBR framework. The weight matrix is taken from the `layer.0.q_proj` layer from the Llama2-7B model. Due to the row-wise decoupling design in OBR, we visualize the distribution of the first row here and give full matrix visualization in Appendix D. The $x$-axis represents the $C_{in}$ channel index, and the $y$-axis denotes the absolute value of weight elements.

results suggest that OBR can potentially serve as a generic post-processing enhancement for existing pruning algorithms to improve performance without retraining.

**OBR for Quantization Only.** We further apply the proposed OBR to a pure quantization-based compression scenario. Specifically, similar to the process described in Sec. 4.3, we first redistribute the rotated weights using OBR compensation to prepare weights more suitable for subsequent quantization. Then, we use the RTN quantizer to obtain low-bit weights. We compare this variant with the baseline that directly applies RTN quantization to the

Table 7: Results of OBR for RTN quantizer in quantization-only tasks.

| Methods | W-A-KV | wiki2↓ | 0-shot↑ |
|---|---|---|---|
| Floating-point | 16-16-16 | 5.47 | 70.47 |
| GPTQ | 4-4-4 | 6.33 | 66.09 |
| RTN | 4-4-4 | 9.04 | 60.10 |
| OBR+RTN | 4-4-4 | 6.87 | 63.98 |

rotated weights without OBR. The results are shown in Tab. 7. As can be seen, the compensation from OBR significantly improves RTN quantization, *e.g.*, 2.17 reduction in perplexity and a 3.88% gain in zero-shot accuracy. Although OBR is not specifically designed for quantization, OBR+RTN still achieves comparable results to GPTQ with a 0.54 perplexity gap. These results demonstrate the potential of our proposed method in quantization-only tasks.

**Illustrative Visualization of OBR.** In Fig. 6, we visualize the weight distribution at different stages of the proposed OBR pipeline. The $\Delta\mathbf{w}^{prune}$ can effectively recover the information loss caused by pruning while preserving the original sparsity. Moreover, the compensation $\Delta\mathbf{w}^{OBR}$ does not introduce additional outliers, and this flat distribution facilitates the subsequent quantization process. At last, the magnitude of the compensation introduced by OBR is comparable to that of the original weights, indicating that our OBR compensation is not noise but structured information capable of restoring the knowledge lost during compression.

## 6 CONCLUSION

In this work, we propose Optimal Brain Restoration (OBR), a unified framework that jointly performs pruning and quantization by computing an optimal compensation to reconcile the conflicting requirements of different compression methods. We begin by formulating a second-order Hessian-based objective that minimizes downstream task degradation. To make the optimization tractable, we introduce a row-wise decoupling approximation. Furthermore, we develop group error compensation, which redistributes compression-induced errors through a closed-form solution. By aligning the weight distribution with the distinct demands of each compression technique, OBR is among the first methods to support INT4 quantization combined with 50% sparsity for LLMs. Experimental results demonstrate that our approach significantly outperforms existing methods and achieves up to $4.72\times$ practical speedup over the FP16-dense baseline.

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

# APPENDIX

## A SUMMARY OF OBR ALGORITHM

In Algo. 1, we provide a detailed pseudocode to illustrate the process of obtaining joint low-bit and sparse LLM weights in the proposed OBR framework.

---

**Algorithm 1** Optimal Brain Restoration (OBR)

---

**Input:** Hadamard rotated weight matrix $\mathbf{W} \in \mathbb{R}^{C_{out} \times C_{in}}$, Hessian approximation $\mathbf{H} \in \mathbb{R}^{C_{in} \times C_{in}}$, partitioning ratio $\alpha$.
**Output:** Low-bit and sparse weight $\hat{\mathbf{W}} \in \mathbb{Z}^{C_{out} \times C_{in}}$.

// Step1 Pruning
$\mathbf{M} \in \{0, 1\} = \texttt{prune}(\mathbf{W})$
$\mathbf{W}^{prune} \leftarrow \mathbf{W} \odot \mathbf{M}$
// Step2 OBR compensation
Initialize $\Delta \mathbf{W}^{OBR}$ as zero matrices in $\mathbb{R}^{C_{out} \times C_{in}}$
**for** $c = 1 \ldots C_{out}$ **do**
  // OBR for pruning
  $R_1 \leftarrow \{i \mid \mathbf{M}_{c,i} = 1\}, \quad E_1 \leftarrow \{j \mid \mathbf{M}_{c,j} = 0\}$
  $\mathbf{b}_1 \leftarrow \mathbf{H}_{R_1 E_1} \cdot \mathbf{W}_{c,E_1}^{\top}$
  $\Delta \mathbf{w}_{R_1}^{prune} \leftarrow -\mathbf{H}_{R_1 R_1}^{-1} \mathbf{b}_1$
  $\bar{\mathbf{w}} \leftarrow \mathbf{W}_{c,R_1}^{prune} + \Delta \mathbf{w}_{R_1}^{prune}$
  // OBR for quantization
  $\mathbf{e}^{quant} \leftarrow \bar{\mathbf{w}} - \texttt{quantize}(\bar{\mathbf{w}})$
  $t \leftarrow \lfloor \alpha \cdot |R_1| \rfloor$
  $E_2 \leftarrow \{r_1, \ldots, r_t\}, \quad R_2 \leftarrow \{r_{t+1}, \ldots, r_{|R|}\}$
  $\mathbf{b}_2 \leftarrow \mathbf{H}_{R_2 E_2} \cdot \mathbf{e}_{E_2}^{quant}$
  $\Delta \mathbf{w}_{R_2}^{quant} \leftarrow -\mathbf{H}_{R_2 R_2}^{-1} \mathbf{b}_2$
  // Compensation Gathering
  $\Delta \mathbf{W}_{c,R_1}^{OBR} += \Delta \mathbf{w}_{R_1}^{prune}$
  $\Delta \mathbf{W}_{c,R_2}^{OBR} += \Delta \mathbf{w}_{R_2}^{quant}$
**end for**
$\mathbf{W}^{quant} \leftarrow \mathbf{W}^{prune} + \Delta \mathbf{W}^{OBR}$
// Step3 Quantization
$\hat{\mathbf{W}} \leftarrow \texttt{quantize}(\mathbf{W}^{quant})$

---

## B COEXISTENCE OF QUANTIZATION AND PRUNING.

A key motivation behind the proposed OBR is the compatibility of low-bit quantization and sparsity in the Hadamard-rotated LLMs. In this section, we provide empirical evidence to justify this motivation. Specifically, we visualize the sparsity distribution of Llama2-7B and Qwen2.5-7B models quantized by different rotation frameworks, *i.e.*, QuaRot (Ashkboos et al., 2024), SpinQuant (Liu et al., 2024), and FlatQuant (Sun et al., 2024). Fig. 7 offers the results. Interestingly, even without any explicit pruning operations, the quantized LLMs inherently exhibit non-trivial sparsity. For instance, Llama2-7B with QuaRot reaches an average sparsity of 14.28%. Based on the observation of this coexistence, we design our OBR to achieve more aggressive LLM compression.

## C MORE EXPERIMENTS

**Comparison with BitNet.** BitNet-2B-4T (Ma et al., 2025) is a recently proposed 1.58-bit LLM that is trained from scratch to achieve aggressive compression with strong performance. In this section, we give a brief comparison between the BitNet-2B-4T model and Qwen2.5 compressed

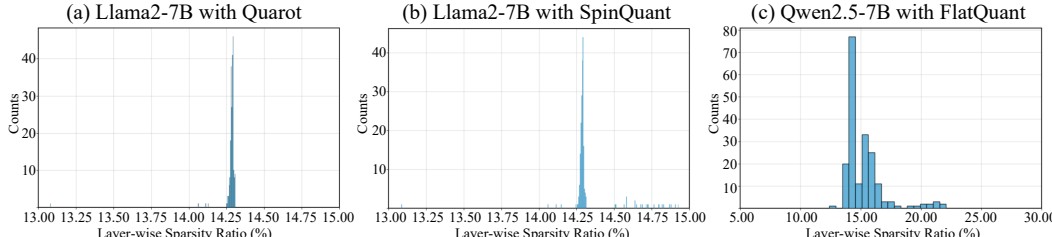

Figure 7: Distribution of layer-wise sparsity across LLMs under different rotation methods. All models are quantized with W4A4KV4 RTN quantizer.

Table 8: Comparison between BitNet-2B-4T and our OBR compressed Qwen2.5-Instruct models.

| methods | quantization | sparisty | PIQA | BoolQ | HellaSwag | ARC-E | ARC-C | WinoGrande | Avg. | Wiki2 |
|---|---|---|---|---|---|---|---|---|---|---|
| BitNet-2B-4T | W1.58A8KV16 | 0% | 76.55 | 80.43 | 68.39 | 74.66 | 49.40 | 72.22 | 70.27 | 13.67 |
| Qwen2.5-1.5B + OBR | W4A8KV16 | 50% | 68.99 | 66.88 | 52.68 | 62.50 | 35.24 | 60.77 | 57.84 | 15.06 |
| Qwen2.5-1.5B + OBR | W4A4KV4 | 50% | 67.25 | 68.01 | 51.18 | 56.99 | 32.94 | 55.96 | 55.38 | 14.92 |
| Qwen2.5-3B + OBR | W4A8KV16 | 50% | 74.05 | 77.19 | 62.86 | 60.06 | 41.30 | 62.90 | 63.06 | 11.07 |
| Qwen2.5-3B + OBR | W4A4KV4 | 50% | 72.14 | 76.67 | 60.43 | 60.69 | 41.13 | 65.59 | 62.77 | 11.79 |

Table 9: Ablation experiments on other calibration dataset. We change the calibration set to the C4 (Raffel et al., 2020) dataset for the generation of activation statistics and keep other setups the same.

| dataset | method | Llama2-7B | | Llama2-13B | | Llama3-8B | |
|---|---|---|---|---|---|---|---|
| | | perplexity↓ | 0-shot↑ | perplexity↓ | 0-shot↑ | perplexity↓ | 0-shot↑ |
| wikitext2 | SparseGPT+GPTQ | 12.94 | 51.57 | 7.89 | 60.74 | 16.40 | 53.77 |
| | Ours_RTN | 9.23 | 56.49 | 7.29 | 62.37 | 14.47 | 54.40 |
| | Ours_GPTQ | 8.40 | 53.45 | 7.06 | 62.60 | 13.92 | 55.16 |
| c4 | SparseGPT+GPTQ | 18.36 | 51.18 | 9.69 | 60.48 | 23.02 | 53.87 |
| | Ours_RTN | 10.74 | 58.00 | 8.74 | 62.88 | 18.23 | 56.02 |
| | Ours_GPTQ | 10.40 | 57.95 | 8.22 | 63.16 | 17.90 | 57.12 |

using our OBR. As shown in Tab. 8, our post-training method achieves comparable performance. To be specific, Qwen2.5-3B+OBR (W4A4KV4+50%Sparsity) achieves better perplexity on WikiText2 and comparative performance on zero-shot accuracy. It should be noted that the performance of OBR can be further boosted when future, more advanced base LLMs are proposed. Moreover, the resulting W4A4KV4+50% sparse LLMs can be seamlessly deployed, such as in NVIDIA Ampere and Hopper, whereas BitNet requires specially designed kernels and customized implementations. At last, our method provides stronger generalization and flexibility. BitNet currently offers only one model size and typically requires training from scratch, which is computationally expensive and impractical for users with domain-specific or confidential data. In contrast, our OBR framework is a general post-training compression approach that can be directly applied to existing models of different sizes, enabling users to efficiently adapt their own LLMs without re-training.

**Ablation on other Calibration Set.** In the proposed OBR, we use the WikiText-2 (Merity et al., 2016) dataset to obtain activation statistics. To further verify the robustness across different calibration sets, we additionally experiment with the C4 (Raffel et al., 2020) dataset for calibration. The results are shown in Tab. 9. As can be seen, when switching to the C4 dataset, all compared methods suffer a slight performance degradation on WikiText perplexity due to the train-test shift. However, models calibrated with C4 achieve better results on zero-shot tasks, and this advantage is more pronounced with our OBR. For example, in the Llama3-8B experiment with C4, SparseGPT+GPTQ achieves only a 0.1% accuracy improvement, whereas the proposed OBR_GPTQ delivers a 1.96% gain. Moreover, both OBR_RTN and OBR_GPTQ consistently outperform the SparseGPT+GPTQ baseline across all calibration sets and base models under the same compression settings. The above results demonstrate the generalization of our method under other calibration sets.

Table 10: Comparison of perplexity score on WikiText2 and accuracy on zero-shot common sense reasoning tasks using the rotation matrix from FlatQuant (Sun et al., 2024).

| Model | Method | #Bits (W-A-KV) | Sparsity ratio | PIQA (↑) | BoolQ (↑) | HellaS. (↑) | Arc-e (↑) | Arc-c (↑) | WinoG. (↑) | Avg. (↑) | Wiki2 (↓) |
|---|---|---|---|---|---|---|---|---|---|---|---|
| Llama2-7B | Floating-point | 16-16-16 | 0% | 79.11 | 77.71 | 76.02 | 74.49 | 46.33 | 69.14 | 70.47 | 5.47 |
| | FlatQuant(quant-only) | 4-4-4 | 0% | 77.48 | 74.62 | 73.64 | 72.56 | 43.00 | 68.27 | 68.26 | 5.79 |
| | FlatQuant(quant-only) | 3-4-4 | 0% | 75.68 | 73.94 | 69.44 | 67.85 | 40.96 | 64.17 | 65.34 | 6.74 |
| | SparseGPT+GPTQ | 4-4-4 | 50% | 73.56 | 50.40 | 65.36 | 61.11 | 34.73 | 62.75 | 57.99 | 7.75 |
| | Ours_RTN | 4-4-4 | 50% | 74.32 | 72.91 | 65.88 | 64.94 | 37.88 | 65.82 | 63.62 | 6.88 |
| | Ours_GPTQ | 4-4-4 | 50% | 74.37 | 71.41 | 65.92 | 64.06 | 38.82 | 66.38 | 63.49 | 6.87 |
| Llama2-13B | Floating-point | 16-16-16 | 0% | 80.52 | 80.55 | 79.37 | 77.48 | 49.15 | 72.14 | 73.20 | 4.88 |
| | FlatQuant(quant-only) | 4-4-4 | 0% | 79.00 | 79.39 | 77.44 | 76.47 | 48.72 | 70.17 | 71.86 | 5.11 |
| | FlatQuant(quant-only) | 3-4-4 | 0% | 78.56 | 78.04 | 75.35 | 70.66 | 44.97 | 70.09 | 69.61 | 5.70 |
| | SparseGPT+GPTQ | 4-4-4 | 50% | 75.90 | 74.53 | 69.81 | 68.86 | 40.19 | 67.09 | 66.06 | 6.13 |
| | Ours_RTN | 4-4-4 | 50% | 76.66 | 73.94 | 71.44 | 71.30 | 42.06 | 68.27 | 67.27 | 5.84 |
| | Ours_GPTQ | 4-4-4 | 50% | 76.61 | 73.27 | 71.39 | 72.10 | 42.49 | 68.43 | 67.38 | 5.84 |
| Llama3-8B | Floating-point | 16-16-16 | 0% | 80.85 | 80.98 | 79.17 | 77.74 | 53.24 | 73.40 | 74.23 | 6.13 |
| | FlatQuant(quant-only) | 4-4-4 | 0% | 79.33 | 79.36 | 76.64 | 75.21 | 48.46 | 72.06 | 71.84 | 6.97 |
| | FlatQuant(quant-only) | 3-4-4 | 0% | 75.68 | 69.42 | 71.21 | 67.47 | 39.85 | 67.40 | 65.17 | 9.14 |
| | SparseGPT+GPTQ | 4-4-4 | 50% | 69.97 | 74.95 | 63.59 | 57.03 | 34.64 | 65.19 | 60.89 | 13.32 |
| | Ours_RTN | 4-4-4 | 50% | 74.16 | 77.61 | 66.86 | 68.81 | 40.78 | 0.6661 | 65.80 | 9.12 |
| | Ours_GPTQ | 4-4-4 | 50% | 73.99 | 77.16 | 66.74 | 69.11 | 41.30 | 68.19 | 66.08 | 9.10 |
| Qwen2.5-7B | Floating-point | 16-16-16 | 0% | 80.14 | 85.96 | 79.57 | 76.47 | 51.19 | 69.46 | 73.78 | 8.35 |
| | FlatQuant(quant-only) | 4-4-4 | 0% | 78.13 | 85.87 | 78.48 | 77.23 | 51.02 | 68.82 | 73.25 | 8.40 |
| | FlatQuant(quant-only) | 3-4-4 | 0% | 73.23 | 82.20 | 74.51 | 69.78 | 48.29 | 63.06 | 68.51 | 10.08 |
| | SparseGPT+GPTQ | 4-4-4 | 50% | 73.56 | 83.70 | 68.50 | 68.10 | 42.49 | 64.01 | 66.72 | 14.53 |
| | Ours_RTN | 4-4-4 | 50% | 74.70 | 85.41 | 71.22 | 74.49 | 49.83 | 66.30 | 70.32 | 9.55 |
| | Ours_GPTQ | 4-4-4 | 50% | 76.66 | 85.08 | 70.68 | 74.12 | 50.85 | 67.56 | 70.82 | 9.51 |
| Qwen2.5-32B | Floating-point | 16-16-16 | 0% | 81.39 | 90.54 | 85.25 | 77.02 | 58.62 | 73.16 | 77.66 | 5.32 |
| | FlatQuant(quant-only) | 4-4-4 | 0% | 80.96 | 89.39 | 83.86 | 79.17 | 57.94 | 73.95 | 77.54 | 5.82 |
| | FlatQuant(quant-only) | 3-4-4 | 0% | 78.94 | 87.83 | 81.45 | 74.87 | 54.69 | 67.64 | 74.23 | 6.79 |
| | SparseGPT+GPTQ | 4-4-4 | 50% | 80.20 | 89.94 | 0.7986 | 73.78 | 52.65 | 72.14 | 74.76 | 8.06 |
| | Ours_RTN | 4-4-4 | 50% | 77.86 | 90.00 | 80.00 | 78.45 | 57.17 | 72.77 | 76.04 | 6.81 |
| | Ours_GPTQ | 4-4-4 | 50% | 79.11 | 89.45 | 80.00 | 77.31 | 59.22 | 72.61 | 76.28 | 6.79 |

**Performance on FlatQuant.** In the main paper, we present the application of our OBR on the LLMs rotated by QuaRot (Ashkboos et al., 2024) or SpinQuant (Liu et al., 2024). To further evaluate the generalization ability of our method on other Hadamard rotation frameworks, we additionally include the comparison results with the FlatQuant (Sun et al., 2024) method. The experimental results are shown in Tab. 10. As can be observed, OBR continues to deliver strong performance compared to the SparseGPT+GPTQ baseline across various base models. Interestingly, comparing with QuaRot and SpinQuant, when using a stronger rotation matrix from FlatQuant, the W4A4KV4 + 50% sparsity LLMs using our OBR can achieve performance on par with their FP16 counterparts. For example, the perplexity gap on Llama2-7B is merely 1.4, compared with the gap of 2.93 in QuaRot. This result further indicates the potential that our OBR can scale in parallel with a more advanced rotation framework.

**Results on Qwen Families.** In this section, we take Qwen2.5-Instruct (7B/32B) as a representative to demonstrate the generalization capability of the proposed OBR on other LLMs. The experimental results are presented in Tab. 10. Given Qwen as the base models, OBR consistently outperforms other strong baselines across different scales. For instance, OBR_RTN surpasses SparseGPT+GPTQ by 4.98 perplexity on the Qwen2.5-7B model. In addition, OBR_RTN also outperforms the quantization-only W3A4KV4 baseline by 0.53 perplexity. These results demonstrate the strong generalization ability of the proposed OBR across different LLM families.

**Calibration Time Cost of OBR.** Tab. 11 reports the time cost for compressing models of different scales using OBR. As one can see, for smaller models such as the 7B model, OBR can produce a W4A4KV4 + 50% LLMs in about 2 hours. For even larger models, such as the 70B, the proposed OBR completes in roughly 36 hours. Since our OBR adopts a row-wise decoupling strategy, it requires solving a linear equation for each row, making it slower than SparseGPT+GPTQ. Nevertheless, we emphasize that post-training compression needs to be performed only once per model. As a result, this cost has only minimal impact on large-scale deployment. Moreover, the promising performance of OBR against other baselines under aggressive compression further justifies its advantages.

Table 11: Calibration time results for Llama model family. The reported times correspond to QuaRot (Ashkboos et al., 2024) rotation on a single A100 GPU.

| Llama family | 2-7B | 2-13B | 2-70B | 3-8B | 3-70B |
|---|---|---|---|---|---|
| SparseGPT+GPTQ | 45min | 54min | 1h53min | 48min | 2h9min |
| OBR_RTN | 2h10min | 4h12min | 35h30min | 2h30min | 35h28min |
| OBR_GPTQ | 2h18min | 4h30min | 35h45min | 2h40min | 35h47min |

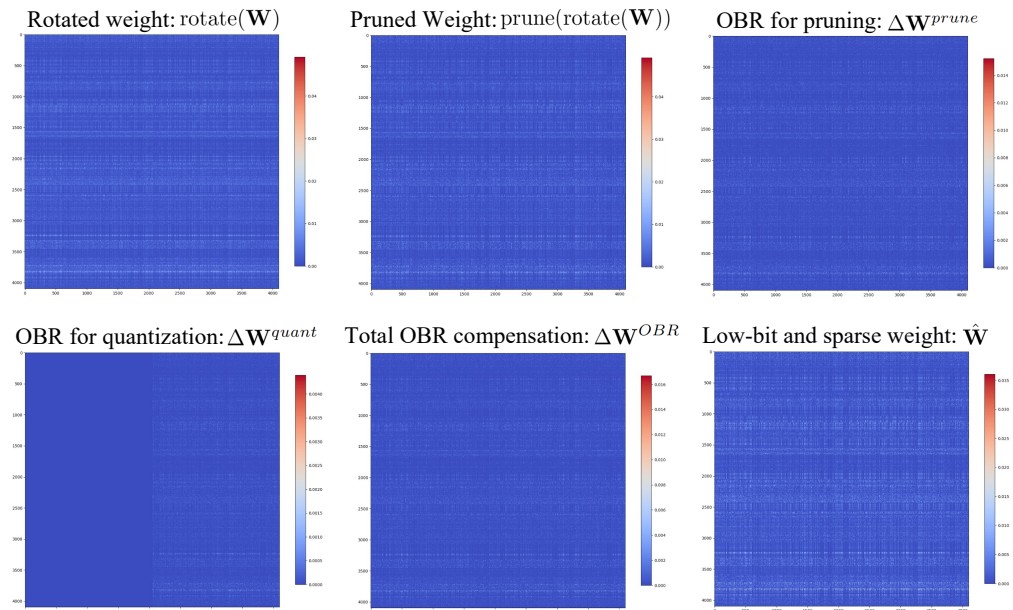

Figure 8: Visualization of the full weight matrix at different stages in the proposed OBR pipeline. The $x$-axis corresponds to the $C_{in}$ dimension, and the $y$-axis is the $C_{out}$ dimension. The weight matrix is taken from the `layer.0.q_proj` layer from the Llama2-7B model, and absolute values are used to enhance visual clarity.

## D   MORE VISUALIZATION

In Fig. 8, we present visualizations of the full weight matrices at different stages of OBR processing. It can be observed that the rotated weight matrix inherently exhibits strong row-wise independence, as indicated by the similarity patterns across rows in $\mathrm{rotate}(\mathbf{W})$. Moreover, the compensation terms $\Delta\mathbf{W}^{prune}$ and $\Delta\mathbf{W}^{quant}$ produced by OBR clearly contain useful information, since they share a similar magnitude with the $\mathrm{prune}(\mathrm{rotate}(\mathbf{W}))$. Therefore, if the OBR compensation were merely noise, perturbations of this magnitude would lead to significant errors. In addition, the overall compensation $\Delta\mathbf{W}^{OBR}$ also demonstrates row-wise independence, where some rows have large magnitudes while others have small ones, yet column dimensions instead exhibit similar patterns. This observation further justifies our proposed row-wise decoupling strategy.

## E   LIMITATION AND FUTURE WORK

While the proposed OBR can effectively redistribute weights to reconcile the differing distributional requirements of quantization and pruning, there are several avenues for further improvement. First, OBR relies on a row-wise decoupling strategy to estimate the full Hessian. This approximation renders the original objective tractable, but it requires solving a linear system for each row of the weight matrix. Although this overhead is acceptable in model compression tasks, where the compression algorithm needs to run only once, further accelerating the compression process for large-scale LLMs remains meaningful. Second, the current implementation of OBR treats the pruning

mask and quantization rotation matrix as fixed given inputs. However, recent quantization studies (Liu et al., 2024; Sun et al., 2024) suggest that introducing gradient-based optimization can further boost performance. Thus, designing learnable pruning masks and rotation matrices compatible with our OBR framework could lead to additional gains. Third, although OBR significantly outperforms individual compression methods under equivalent sub-4-bit settings, its advantage narrows at higher bit-widths, where standalone methods have not yet reached their performance limits. Developing more advanced algorithms to maintain superior performance across various bit-widths is also a promising direction, and we leave it for future work.

