# OpenReview forum: "Optimal Brain Restoration for Joint Quantization and Sparsification of LLMs"
_ICLR.cc/2026/Conference — ICLR 2026 Poster_

### Official Review · Reviewer_Y5Wm · 2025-10-22

**Soundness:** 1
**Presentation:** 1
**Contribution:** 1
**Rating:** 2
**Confidence:** 4

**Summary:**

The key idea is to model the effect of weight perturbations due to pruning + quantization via a second-order Hessian objective, approximate it tractably (row-wise decoupling, approximate Hessian via Fisher × identity factorization, then derive a closed‐form compensation term that “restores” performance by shifting information from more error‐sensitive weights to more robust ones.

**Strengths:**

Model compression for LLMs is a hot area and bridging quantization + pruning is of high practical relevance

**Weaknesses:**

1 The general idea of combining quantization + pruning is not new (the authors themselves reference earlier works on small networks like DJPQ (Wang et al., 2020) and OBQ (Frantar & Alistarh, 2022) for the joint problem).

2 The Hessian‐based analysis for pruning quantization has precedent (e.g., OBD/OBQ for pruning, Fisher/Hessian analyses, quantization with perturbation bounds). The authors are extending to LLM scale, but one might argue that the core conceptual novelty is limited: the compensation term is derived, but the big leap is engineering (applying it to huge models) rather than algorithmic novelty.

3 Some sections feel dense and hard to follow

**Questions:**

1 For pruning, you adopt the mask from WANDA as default. Have you tested OBR using other pruning algorithms

---

> ### Author Response · Authors · 2025-11-20
> **Response to Reviewer Y5Wm [part1]**
>
> We would like to sincerely thank Reviewer Y5Wm for providing a detailed review and insightful suggestions.
>
> > **W1:** Novelty Justification on Joint Quantization and Pruning
>
> Thanks for the reviewer's valuable suggestion. Both the DJPQ, OBQ and our OBR aim to achieve joint quantization and pruning, but they have several essential differences.
>
> 1) The motivation for joint quantization and pruning in our work is different from previous methods. As shown in the compression-scaling curves in Fig.1, we are the first to demonstrate that further LLM compression requires the combination of quantization and pruning. We also empirically verify their compatibility. For example, we find a quantized LLM naturally exhibits 14.28% sparsity. In contrast, previous methods do not specifically study joint compression strategies in the LLM context.
>
> 2) The technical details are different. DJPQ adopts a gradient-based approach, which is impractical for large-scale LLM tuning. OBQ designs pruning and quantization separately, without considering their relations. In contrast, our OBR unifies pruning and quantization within a single framework (as shown in Sec. 4.2), and we are the first to derive a closed-form solution, which is more efficient than previous iterative methods.
>
> 3) In practice, our method is also the first to achieve aggressive LLM compression with good performance. We compare our OBR with the SOTA method JSQ under aggressive compression setups on the Llama2-7B model. Results are shown below, which demonstrates the significant advantages of our proposed method.
>
> | methods | setup                        | PIQA  | BoolQ | HellaS. | Arc-e | Arc-c | WinoG. | Avg  acc  | wiki2  ppl |
> |---------|------------------------------|-------|-------|---------|-------|-------|--------|-------|--------|
> | JSQ     | W4A8KV16+50% sparse          | 74.54 | 70.03 | 66.33   | 65.95 | 38.82 | 64.09  | 63.29 | 9.53   |
> | Ours    | W4A8KV8+50% sparse           | 73.45 | 71.93 | 64.90   | 64.73 | 37.97 | 67.40  | 63.40 | 6.87   |
> | JSQ     | W4A4KV16+50% sparse          | 49.24 | 38.26 | 25.77   | 25.42 | 26.37 | 48.15  | 35.53 | 6674.57|
> | Ours    | W4A4KV4+50% sparse           | 68.93 | 67.31 | 58.22   | 55.93 | 34.22 | 61.48  | 53.45 | 8.40   |
>
>
> > **W2:** Comparison with Previous Hessian‐based Methods
>
> We would like to clarify that both OBD/OBQ and our proposed OBR use Hessian-based analysis, but their specific algorithms are significantly different.
>
> 1) We are the first to derive a closed-form solution under the Hessain framework through approximating the objective function. Specifically, we introduce row-wise decoupling and curvature-matrix approximation to derive this close-form optimal compensation. In contrast, previous frameworks require multiple iterative steps.
>
> 2) Experimentally, we also include SparseGPT+GPTQ in our comparison, a representative method leveraging the OBQ framework at LLM scale. The results are shown below, which shows our OBR method achieves significant performance advantage against other Hessian-based approaches.
>
> 3) The novelty of our algorithm is also well recognized by other reviewers. As Reviewer ruw5 and Reviewer vBwQ have commented,  our OBR is "Sound theoretical development" and  "systematically extends the classic OBD/OBQ theory, generalizing it from single-task scenarios to joint compression settings."
>
>
> |          methods         |            | Zero-shot acc | wiki2-perplexity |
> |-------------------|-------------------|---------------|------------------|
> | Llama2-7B SparseGPT+GPTQ |                   | 57.99        | 7.75            |
> | Llama2-7B Ours    |                   | 63.49        | 6.87            |
> | Llama2-13B SparseGPT+GPTQ |              | 66.06        | 6.13            |
> | Llama2-13B Ours   |                   | 67.38        | 5.84            |
> | Qwen2.5-7B SparseGPT+GPTQ |             | 66.72        | 14.53           |
> | Qwen2.5-7B Ours   |                   | 70.82        | 9.51            |
> | Qwen2.5-32B SparseGPT+GPTQ |            | 74.76        | 8.06            |
> | Qwen2.5-32B Ours  |                   | 76.28        | 6.79            |

---

> ### Author Response · Authors · 2025-11-20
> **Response to Reviewer Y5Wm [part2]**
>
> > **Q1:** Compatibility with Different Pruning Algorithms in OBR
>
> The required experiments are given in Tab.5 of the main text. To make this clearer, we also present the results here.
>
> | Pruning algo.     | PIQA  | BoolQ | HellaSwag | ARC-E | ARC-C | WinoGrande | AVG accuracy | wiki2-ppl |
> |-------------------|-------|-------|-----------|-------|-------|------------|--------------|-----------|
> | Magnitude pruning | 69.53 | 68.29 | 57.19     | 53.11 | 30.89 | 60.06     | 56.51        | 8.92      |
> | SparseGPT pruning | 69.75 | 66.67 | 55.68     | 53.24 | 30.72 | 56.67     | 55.45        | 9.28      |
> | WANDA pruning     | 68.93 | 67.31 | 58.22     | 55.93 | 34.22 | 61.48      | 53.45        | 8.40      |
>
> The results above demonstrate that our OBR can be potentially combined with a variety of popular pruning methods while still achieving strong performance. Moreover, as shown in Tab.1 in the main paper, our OBR is also compatible with many quantization schemes, verifying the generality of the proposed OBR.

---

### Official Review · Reviewer_vBwQ · 2025-10-31

**Soundness:** 3
**Presentation:** 3
**Contribution:** 3
**Rating:** 6
**Confidence:** 5

**Summary:**

This paper proposes Optimal Brain Restoration (OBR), a general training-free framework for jointly achieving quantization and sparsification of Large Language Models (LLMs). Its core motivation stems from the fundamental conflict between quantization and pruning: quantization prefers compact weight distributions (to suppress outliers) while pruning favors high-variance distributions (to identify important weights). OBR reconciles these conflicting objectives by introducing a Hessian-based error compensation mechanism between pruning and quantization, which "transfers" the information lost due to pruning or quantization distortion to more robust weights. This method transforms the original second-order optimization problem into a closed-form solution via row-wise decoupling and group error compensation, making it efficient and interpretable. Experiments show that OBR achieves LLM compression of W4A4KV4 + 50% sparsity, incurring only a 1.4 perplexity increase on Llama2-7B while delivering 4.72× inference speedup and 6.4× memory reduction.

**Strengths:**

1.Applying Classical Hessian Optimization to Modern LLM Joint Compression Scenarios: Closed-Form Solution Design Balances Efficiency and Performance

2.OBR functions as a post-processing module, compatible with various pruning methods, quantizers, and rotation schemes.

3.This work systematically extends the classic Optimal Brain Damage/Surgeon (OBD/OBQ) theory, generalizing it from single-task scenarios to joint compression settings.

4.This work benchmarked the latency, FLOPs, and TOPS of INT4 2:4 sparse GEMM on the A100 GPU and implemented a custom CUTLASS kernel.

**Weaknesses:**

1.The calibration phase requires solving a system of linear equations for each row, resulting in significant time overhead during calibration.

2.The partitioning ratio α in the quantization compensation phase lacks theoretical grounding. Although validated through ablation studies, the optimal α may vary across layers and models.

3.OBR’s calibration data sensitivity is insufficiently analyzed—it is only evaluated within general-domain text corpora (WikiText2 and C4) and lacks cross-domain robustness experiments, making it difficult to guarantee the stability of its compensation mechanism in real-world, domain-specific deployment scenarios.

4.The inter-row decoupling hypothesis, due to ignoring the correlations between channels, may lead to underestimation of error propagation and accumulation of non-orthogonal errors in LLM compression.

**Questions:**

1.Table 6 demonstrates that employing different α values across various models can lead to superior performance. Is it feasible to design an adaptive α strategy?

2.OBR’s compensation relies on the calibration dataset; if the calibration data distribution differs significantly from that of the target downstream task (e.g., calibrating on general-domain text but deploying on mathematical reasoning), could the compensation amplify errors rather than mitigate them?

3.Is the row-wise decoupling assumption in Section 4.1 of the paper reasonable? Could the authors provide an ablation study comparing the error accumulation under the decoupling assumption versus partially coupled alternatives?

4.The paper compares OBR against baselines such as SparseGPT+GPTQ, but does not include comparisons with other joint compression methods like JSQ. Under the W4A4KV4 + 50% sparsity setting, is OBR’s advantage over JSQ and similar methods truly significant?

---

> ### Author Response · Authors · 2025-11-20
> **Response to Reviewer vBwQ  [Part1]**
>
> Many thanks to Reviewer vBwQ for providing a detailed review and questions.
>
> > **W1:** Discussion on Calibration Time Cost
>
> Many thanks for your insightful question. As presented in Table 11 in Appendix C (we also provide the same table below), our OBR takes about 2 hours for a 7B model and about 4h for a larger 13B model.
>
> | Llama family | 2-7B     | 2-13B    | 3-8B     |
> |--------------|----------|----------|----------|
> | SparseGPT+GPTQ | 45min    | 54min    | 48min    |
> | OBR_RTN      | 2h10min  | 4h12min  | 2h30min  |
> | OBR_GPTQ     | 2h18min  | 4h30min  | 2h40min  |
>
> It is worth noting that our calibration time can actually be significantly reduced through CUDA parallelization, since each channel is solved independently, potentially achieving up to ~4096× speedup. Considering that in most cases an LLM only needs to be compressed once for inference, the time cost of the current naive implementation is acceptable. Moreover, aggressive LLM compression such as W4A4KV4 + 50% sparsity inevitably requires more computation for better performance. The additional calibration time of OBR consequently brings substantial improvements compared to the SparseGPT+GPTQ baseline.
>
> > **W2&Q1:** Theoretical Proof and Adaptive Design for Partition ratio $\alpha$
>
> Intuitively, if $\alpha$ is too small, there are not enough elements in the $E$ set for OBR to compensate, resulting in a large total quantization error. Conversely, if $\alpha$ is too large, each element in the $R$ set bears too much compensation, which also increases the total error. Therefore, choosing $\alpha = 0.5$ is an intuitive compromise.
>
> Mathematically, assume that the error of element at position $i$ follows an uniform distribution $e_i \sim U(-u, u)$, where $i=1,2,...,N$. Then, the $(1-\alpha)N$ elements in the $R$ set need to share the total error from $\alpha N$ elements in the $E$ set:
> $e_{total} = \sum_{i=0}^{\alpha N} e_i$, which has an upper bound of $\alpha N u$.
>
> On one hand, we need to ensure that the average compensation burden per element in $R$ is bounded, i.e.,
> $\frac{\alpha N u}{(1-\alpha)N} \le u$, which leads $\frac{\alpha}{1-\alpha} \le 1$ and thus $\alpha \le 0.5$.
>
> On the other hand, we want to include as many elements as possible in the $E$ set for compensation, which favors a larger $\alpha$. Considering these two factors,  $\alpha = 0.5$  achieves the optimal balance.
>
> However, in real data distribution, the quantization error $e_i$ may not perfectly follow the uniform assumption. To make our OBR more robust to the real data, we follow reviewer's suggestion and further design an adaptive $\alpha$ strategy for each channel. Since $\alpha$ is intended to compensate for quantization-induced errors, we can adaptively determine the retain set R2 and eviction set E2 based on the magnitude of the quantization error $|\bar{w} − \mathrm{quant}(\bar{w})|$. Technically, we use the interquartile range to identify outlier elements to form E2, and assign the remaining elements to R2. In this way, the $\alpha=|E2|/(|E2|+|R2|)$ thus becomes adaptive. The results using the FlatQuant rotation scheme with Llama2-7B are as follows, showing that an adaptive group-partition strategy is more beneficial for performance.
>
> | setup                  | PIQA  | BoolQ | HellaS. | Arc-e | Arc-c | WinoG. | Avg   | wiki2 |
> |------------------------|-------|-------|---------|-------|-------|--------|-------|-------|
> | OBR baseline           | 74.37 | 71.41 | 65.92   | 64.06 | 38.82 | 66.38  | 63.49 | 6.87  |
> | + adaptive alpha$      | 74.37 | 73.03 | 65.98   | 64.98 | 39.71 | 66.54  | 64.10 | 6.77  |

---

> ### Author Response · Authors · 2025-11-20
> **Response to Reviewer vBwQ [Part2]**
>
> > **W3&Q2:** Experiments on Calibration Data Sensitivity
>
> Following your suggestion, we conduct the evaluation on LongBench. This benchmark contains multiple tasks designed to assess an LLM’s long-context understanding capabilities, with some subtasks having an average length of up to 20K tokens. We compare the proposed OBR with SparseGPT+GPTQ, both applied to the Qwen2.5-7B model calibrated on the WikiText-2 dataset. The compression setting is W4A4KV4 + 50% sparsity. This experiment validates the robustness of our method under domain shifts in context length.
>
> | Method        | Single-Document QA (%) | Multi-Document QA (%) | Summarization (%) | Few-shot Learning (%) |
> |-----------------------|-------------------------|------------------------|-------------------|-----------------------|
> | QWen2.5-7B-FP         | 40.57                   | 44.32                  | 26.48             | 66.16                 |
> | SparseGPT+GPTQ        | 14.21                   | 8.44                   | 24.73             | 57.50                 |
> | Ours                  | 28.55                   | 32.65                  | 26.54             | 62.17                 |
>
>
> Under aggressive LLM compression, our method achieves more robust performance compared to SparseGPT+GPTQ.
>
> Furthermore, we provide additional comparisons on the GSM8K mathematical reasoning task. To evaluate cross-domain generalization, we directly test the WikiText2-calibrated compressed LLM without any task-specific knowledge. Following the protocol from `lm-eval`, we report results on the Qwen2.5-7B-Instruct model with and w/o CoT (using the prompt “think step by step”), corresponding to the `gsm8k` and `gsm8k-cot` task tags in `lm-eval`, respectively. We compare against the SparseGPT+GPTQ baseline. The results are shown below, which  demonstrates that our method preserves robust performance on mathematical reasoning tasks.
>
> |                | Qwen2.5-7B-FP | SparseGPT+GPTQ | Ours             |
> |----------------|---------------|----------------|------------------|
> | compression setup | W16A16KV16dense    | W4A4KV4+50%sparse    | W4A4KV4+50%sparse |
> | gsm8k           | 42.38%        | 25.77%         | 40.33%           |
> | gsm8k-cot       | 45.86%        | 30.40%         | 44.12%           |
>
>
>
> > **W4&Q3:** Rationale of Row-wise Decoupling
>
> Thank you for your insightful question. In the proposed OBR, channel-wise decoupling occurs along the $C_{out}$ dimension for weight $W \in \mathbb{R}^{C_{out}\times C_{in}}$. Intuitively, when computing $\mathbf{WX}$ in one linear layer, the computations across different $C_{out}$ channels are parallel and independent. For example, reordering each row of $\mathbf{W}$ corresponds to reordering each row of $\mathbf{WX}$, meaning weak correlations across different rows.
>
> As suggested, we further design a partially coupled experiment, which considers the correlations between every two adjacent channels. It is worth noting that our algorithm can in principle be extended to interactions among more channels or even all $C_{out}$ channels provided sufficient computation.  To achieve the interaction between two channels, we need to additionally estimate a $2 \times 2$ correlation matrix between every two channels to approximate the curvature matrix $\mathbf{G}$ in Eq. 5 (recall that $\mathbf{G}$ reduces to an identity matrix under the original independence assumption). Under this setting, the full Hessian matrix changes from original block-diagonal form to new
>
> $\mathbf{H}_{full} = \mathbf{G} \otimes \mathbf{H} \in \mathbb{R}^{2 Cin \times 2Cin}$
>
> where $\otimes$ denotes Kronecker product. The error vector $e$ also becomes the concatenation of the two adjacent channels, with shape [1, 2$C_{in}$]. As can be seen, modeling interactions between two channels introduces significant additional computational overhead due to the enlarged matrix dimensions.
>
> The results are listed as follows, which can observe that there is no significant improvement when using partially coupled strategy but incurring more calibration time costs.
>
> | setup                  | Zero-shot acc | wiki2-perplexity | Calibration time |
> |------------------------|---------------|------------------|------------------|
> | OBR baseline           | 63.49         | 6.87             | 2h18min          |
> | + partially coupled    | 63.82         | 6.89             | 5h54min          |

---

> ### Author Response · Authors · 2025-11-20
> **Response to Reviewer vBwQ [Part3]**
>
> > **Q4:** Comparison with JSQ
>
> Following the reviewer’s suggestion, we supplement additional experiments to compare JSQ under different compression settings. The results on Llama2-7B are shown as follows.
>
>
> | methods | setup                        | PIQA  | BoolQ | HellaS. | Arc-e | Arc-c | WinoG. | Avg acc  | wiki2 ppl |
> |---------|------------------------------|-------|-------|---------|-------|-------|--------|-------|--------|
> | JSQ     | W4A8KV16+50% sparse          | 74.54 | 70.03 | 66.33   | 65.95 | 38.82 | 64.09  | 63.29 | 9.53   |
> | Ours    | W4A8KV8+50% sparse           | 73.45 | 71.93 | 64.90   | 64.73 | 37.97 | 67.40  | 63.40 | 6.87   |
> | JSQ     | W4A4KV16+50% sparse          | 49.24 | 38.26 | 25.77   | 25.42 | 26.37 | 48.15  | 35.53 | 6674.57|
> | Ours    | W4A4KV4+50% sparse           | 68.93 | 67.31 | 58.22   | 55.93 | 34.22 | 61.48  | 53.45 | 8.40   |
>
> It is notable that under W4A8 settings, KV-4bit OBR even outperforms KV-16bit JSQ. Moreover, the JSQ algorithm collapses under aggressive LLM compression settings, i.e., W4A4KV16 + 50% sparsity. Our proposed OBR is among the first to achieve aggressive LLM compression with acceptable performance.

---

### Official Review · Reviewer_ruw5 · 2025-10-31

**Soundness:** 3
**Presentation:** 4
**Contribution:** 3
**Rating:** 6
**Confidence:** 3

**Summary:**

This paper proposes Optimal Brain Restoration (OBR), which reconciles the conflict between pruning and quantization via a second-order approximation and a closed-form group error compensation scheme: errors introduced on pruned/quantized weights are projected—through Hessian sub-blocks—onto the retained weights to minimize accuracy loss under joint compression. On models such as Llama-2/3 and Qwen-2.5, OBR achieves better zero-shot accuracy and WikiText-2 perplexity than strong baselines (e.g., SparseGPT+GPTQ and naïve “quantization+pruning”) under the W4A4KV4 + 50% sparsity setting. For efficiency, the authors implement an INT4 + 2:4 sparse GEMM using NVIDIA CUTLASS and report kernel-level latency/throughput curves (e.g., a 5.9× speedup over dense FP16 and 1.4× over dense INT4 at sequence length 4096).

**Strengths:**

1. Clear and well-justified motivation. The paper precisely frames the problem: quantization and pruning each approach their limits, and naïvely combining them causes mutual interference, motivating a method that explicitly resolves this conflict. It also provides empirical evidence that non-trivial sparsity naturally emerges after quantization, supporting the coexistence and complementarity of the two representations and thus the direction of joint compression.

2. Sound theoretical development. Starting from a second-order (Hessian-based) objective, the paper introduces group error compensation, derives a closed-form optimum, and explains the Hessian’s bridging role in transferring errors—offering good interpretability.

3. Methodological generality. The proposed approach can be combined with popular sparsity/quantization techniques such as SparseGPT, OBR, QuaRot, SpinQuant, and FlatQuant, making it a broadly applicable compression method.

4. Custom kernel implementation. The authors implement dedicated GEMM CUDA kernels with NVIDIA CUTLASS (INT4 with 2:4 sparsity), which strengthens practical usability and reflects substantial engineering effort.

5. Comprehensive setup and strong results. Experiments span multiple sizes of Llama-2/3 and Qwen-2.5, evaluating zero-shot benchmarks and WikiText-2 perplexity, with comparisons against quantization-only and strong prune+quantize baselines (e.g., SparseGPT+GPTQ). The method shows high speedups and substantial memory reduction.

**Weaknesses:**

1. Lack of end-to-end inference metrics. The paper does not report TTFT (time-to-first-token), tokens/s, or throughput under concurrency. Although kernel-level gains are provided, their translation to full-pipeline performance is unclear.


2. Narrow evaluation tasks. The experiments could be broadened to include more task types—such as code generation or long-context workloads—to better demonstrate generality.

**Questions:**

same as the weaknesses.

---

> ### Author Response · Authors · 2025-11-20
> **Response to Reviewer ruw5**
>
> Many thanks to Reviewer ruw5 for providing a detailed review and insightful questions.
>
> > **W1**: Efficiency Comparison on End-to-end Inference Metrics
>
> Following your suggestion, we supplement the end-to-end inference efficiency. We use Llama2-7B as the base model, perform pre-filling with 1024 tokens, and have the model generate an additional 1024 tokens as the response (i.e. a total of 2048 tokens to process). The experiments are conducted on a single A100 GPU, and the results are listed below.
>
>
> | setup        | TTFT    | Tokens/s | Total generation time |
> |--------------|---------|----------|-----------------------|
> | FP16-Dense   | 27.76ms | 37.48    | 27.32s                |
> | INT4-Dense   | 7.13ms  | 145.93   | 7.02s                 |
> | INT4-Sparse  | 5.24ms  | 198.43   | 5.17s                 |
>
> We observe that the proposed joint INT4 + 50% sparse scheme achieves a significant improvement in end-to-end throughput. Specifically, the proposed joint quantization-and-sparsity method achieves a 5.3$\times$ speedup compared to the FP16-dense baseline and a 1.4$\times$ speedup compared to the INT4-dense baseline.
>
> > **W2:** Experiments on Long-context Benchmarks
>
> Following your suggestion, we conduct the evaluation on LongBench. This benchmark contains multiple tasks designed to assess an LLM’s long-context understanding capabilities, with some subtasks having an average length of up to 20K tokens. We compare the proposed OBR with the strong baseline SparseGPT+GPTQ, both applied to the Qwen2.5-7B model calibrated on the WikiText-2 dataset. The compression setting is W4A4KV4 + 50% sparsity. The experimental results are shown below.
>
> | methods            | NrtrvQA | Qasper | MFen | HotpotQA | 2WikiMQA | Musique | GovReport | QMSum | MultiNews | TREC | TriviaQA | SAMSum |
> |--------------------|---------|--------|-------|----------|----------|---------|-----------|--------|------------|------|-----------|---------|
> | QWen2.5-7B-FP      | 26.11   | 43.58  | 52.02 | 57.5     | 44.89    | 30.57   | 32.77     | 22.82  | 23.84      | 66.5 | 87.12     | 44.86   |
> | SparseGPT+GPTQ     | 4.41    | 12.94  | 25.28 | 9.52     | 9.24     | 6.57    | 29.76     | 19.77  | 24.65      | 62.5 | 71.98     | 38.02   |
> | Ours               | 17.89   | 23.11  | 44.65 | 44.35    | 35.06    | 18.55   | 32.99     | 22.02  | 24.62      | 64.5 | 79.34     | 42.68   |
>
> Under aggressive LLM compression, our method achieves more robust performance compared to SparseGPT+GPTQ. It is notable that the performance is expected to further improve if long-context corpora are used as the calibration set.

---

### Official Review · Reviewer_JF8r · 2025-11-01

**Soundness:** 3
**Presentation:** 3
**Contribution:** 3
**Rating:** 6
**Confidence:** 5

**Summary:**

This paper propose Optimal Brain Restoration (OBR), combining the sparse pruning and quantization with error compensation for model compression. Experiments show that OBR incurs only a 1.4 perplexity degradation on Llama2-7B to enable aggressive W4A4KV4 quantization with 50% sparsity, delivering up to 4.72× speedup and 6.4× memory reduction compared to the FP16-dense baseline.

**Strengths:**

A training free method with the combination of pruning and quantization is achieved, by the intervene between pruning and quantization by computing an optimal compensation that minimizes the impact of compression on downstream tasks.

The experiments show SOTA results, indicating the first algorithm to enable W4A4KV4+50% sparsity LLMs, without requiring any additional retraining.

**Weaknesses:**

The sparse pruning can be used based on 4:2 sparsity, the hardware implementation is efficient, but the accuracy drop of W4A4KV4+50% sparsity LLMs is larger than 5%, which hinders the application of this method. The experimental results are based on simple QA tasks, and lack of long context tasks and long-cot tasks, so the results are not convincing.

Also the algorithm should verify the results on the new data formats such as NVFP4 or MXFP4, which are popular on the modern hardware.

**Questions:**

1. Maybe the training aware method by combining the quantization and pruning is more practical, especially for the use of W4A4 + 4:2 sparsity.
2. What's dominant application of 4:2 sparsity? Since few examples of 4:2 sparsity are released in the industry.

---

> ### Author Response · Authors · 2025-11-20
> **Response to Reviewer JF8r [Part1]**
>
> We would like to sincerely thank Reviewer JF8r for providing a detailed review and insightful suggestions.
>
> > **W1.1&Q1:** Improve Performance on 2:4/4:8 Semi-structured Pruning
>
> Following your suggestion, we explore two approaches to improve OBR on semi-structured pruning.
>
> First, we replace the fixed randomized Hadamard rotation with a learnable rotation matrix, which we denote as **trainable_rotation**.
>
> Second, inspired by [A], we further propose a training-free method to improve performance. Specifically, given $\mathbf{W} \in \mathbb{R}^{C_{out}\times C_{in}}$, $\mathbf{X} \in \mathbb{R}^{C_{in}\times L}$, we compute a channel-wise importance score of size $C_{in}$ using $|\mathbf{W}|.sum(dim=0) \cdot |\mathbf{X}|.sum(dim=1) $. We then reorder both W and X along the $C_{in}$ dimension according to this score to manually construct an N:M sparsity pattern with small errors. Since the reordering is performed along the inner dimension of the matrix multiplication, it does not affect the final output. During inference, we only need to reorder W and X using the pre-computed indices, w/o additional overhead. We denote this method as **channel_permute**.
>
> | setup               | N:M sparsity | wiki2-perplexity | Zero-shot acc |
> |---------------------|--------------|------------------|----------------|
> | Llama2-7B-FP16      | -            | 5.47             | 70.46          |
> | Ours (baseline)     | 4:8          | 10.61            | 52.02          |
> | + trainable_rotation| 4:8          | 8.14             | 58.38          |
> | + channel_permute   | 4:8          | 9.51             | 54.65          |
> | Ours (baseline)     | 2:4          | 13.32            | 48.67          |
> | + trainable_rotation| 2:4          | 9.81             | 54.75          |
> | + channel_permute   | 2:4          | 11.74            | 51.30          |
>
> We observe that the learnable rotation method significantly improves performance, achieving a 2.47 perplexity reduction under the 4:8 sparsity setting. In addition, the training-free channel permutation also improves performance by alleviating the pruning-assignment constraints imposed by semi-structured sparsity.
>
> [A] Channel Permutations for N:M Sparsity. NeurIPS 2021
>
>
> > **W1.2:** Comparison on Long Context Tasks
>
> Following your suggestion, we complete the evaluation on LongBench. This benchmark contains multiple tasks designed to assess LLMs’ ability in long-context understanding, with some subtasks having an average length of 20K. We compare the proposed OBR with the strong baseline SparseGPT+GPTQ, both applied to the Qwen2.5-7B model calibrated on the WikiText-2 dataset. The compression setting is W4A4KV4 + 50% sparsity. Results are shown below.
>
> | methods            | NrtrvQA | Qasper | MFen | HotpotQA | 2WikiMQA | Musique | GovReport | QMSum | MultiNews | TREC | TriviaQA | SAMSum |
> |--------------------|---------|--------|-------|----------|----------|---------|-----------|--------|------------|------|-----------|---------|
> | QWen2.5-7B-FP      | 26.11   | 43.58  | 52.02 | 57.5     | 44.89    | 30.57   | 32.77     | 22.82  | 23.84      | 66.5 | 87.12     | 44.86   |
> | SparseGPT+GPTQ     | 4.41    | 12.94  | 25.28 | 9.52     | 9.24     | 6.57    | 29.76     | 19.77  | 24.65      | 62.5 | 71.98     | 38.02   |
> | Ours               | 17.89   | 23.11  | 44.65 | 44.35    | 35.06    | 18.55   | 32.99     | 22.02  | 24.62      | 64.5 | 79.34     | 42.68   |
>
> Under aggressive LLM compression, our method achieves more robust performance compared to SparseGPT+GPTQ. It is notable that the performance is expected to further improve if long-context corpora are used as the calibration set.
>
> Furthermore, we provide additional comparisons on the GSM8K mathematical reasoning task. To evaluate cross-domain generalization, we directly test the WikiText2-calibrated compressed LLM without any task-specific knowledge. Following the protocol from `lm-eval`, we report results on the Qwen2.5-7B-Instruct model with and w/o CoT (using the prompt “think step by step”), corresponding to the `gsm8k` and `gsm8k-cot` task tags in `lm-eval`, respectively. We compare against the SparseGPT+GPTQ baseline. The results are shown below, which  demonstrates that our method preserves robust performance on mathematical reasoning tasks.
>
> | compression setup | Qwen2.5-7B-FP | SparseGPT+GPTQ   | Ours                 |
> |-------------------|---------------|-------------------|-----------------------|
> | W16A16KV16        | W16A16KV16    | W4A4KV4+50%       | W4A4KV4+50%sparse     |
> | gsm8K             | 42.38%        | 25.77%            | 40.33%                |
> | gsm8K-cot         | 45.86%        | 30.40%            | 44.12%                |

---

> ### Author Response · Authors · 2025-11-20
> **Response to Reviewer JF8r [Part2]**
>
> > **W2:** Verification of new Data Formats MXFP4
>
> As per your request, we conduct experiments under the MXFP4 dtype setup. Specifically, we replace the previous integer quantizer with MXFP4 format while keeping all other settings unchanged. In this case, the proposed OBR is used to compensate for the errors introduced by sparsity and the MXFP4 dtype conversion. The experimental results are listed as follows.
>
> | setup       | dtype (W-A-KV) | Sparsity ratio | wiki2-perplexity | zero-shot  acc|
> |-------------|----------------|----------------|------------------|-----------|
> | OBR (baseline) | INT4-INT4-INT4 | 50%            | 9.23             | 56.49     |
> | MXFP4       | MXFP4-MXFP4-MXFP4 | 50%            | 8.39             | 62.22     |
>
> Our experiments show that using MXFP4 yields better performance compared to integer quantization, due to the nonlinearity of MXFP4. Since we currently do not have access to Blackwell GPUs, we are unable to conduct experiments with NVFP4. We will address this in future work.
>
>
> > **Q2:** Industrial Application on 2:4 Sparsity
>
> Thank you for your practical question. Although there are relatively few publicly released 2:4 sparse LLMs, this sparsity pattern is widely used in industry. For instance, NVIDIA’s Ampere and Hopper GPUs provide hardware acceleration for 2:4 structured sparsity through Sparse Tensor Cores, and cuSPARSELt further enables efficient deployment [B]. In addition, inference frameworks such as TensorRT also provide native support for 2:4 sparse weights[C]. Moreover, as shown in this paper, sparsity can potentially be combined with quantization techniques to achieve aggressive LLM compression.
>
> [B] https://developer.nvidia.com/blog/exploiting-ampere-structured-sparsity-with-cusparselt
>
> [C] https://developer.nvidia.com/blog/structured-sparsity-in-the-nvidia-ampere-architecture-and-applications-in-search-engines

---

> > ### Comment · Reviewer_JF8r · 2025-11-24
> >
> > Results seem good！

---

> > > ### Author Response · Authors · 2025-11-25
> > > **Thanks for your response and raising score.**
> > >
> > > Thank you for your prompt response and valuable comments on our paper, which have been of great help to us. We promise to follow your suggestion to improve our manuscript's quality in the revised revision.

---

### Author Response · Authors · 2025-11-20
**Summary of Revisions and Global Response**

We sincerely thank all the reviewers for their insightful reviews and valuable comments, which are instructive for us to improve our paper further.

In this paper, we dive into the combination of quantization and pruning for more aggressive LLM compressions. We propose OBR (Optimal Brain Restoration) which compute an optimal compensation to reconcile the conflicting requirements of different compression methods. Experimentally, OBR achieves **SoTA performance** on extreme W4A4KV4 + 50% sparse LLM compression scenarios, delivering up to **4.72× speedup** and **6.4× memory reduction** compared to the FP16-dense baseline.

The reviewers generally held positive opinions of our paper. We are encouraged that they find our method:

- **"the first algorithm to enable W4A4KV4+50% sparsity LLMs"** (Reviewer JF8r)

- **"Clear and well-justified motivation"** (Reviewer ruw5)

- **"Sound theoretical development."** (Reviewer ruw5, Reviewer vBwQ)

- **"compatible with various pruning methods, quantizers, and rotation schemes."** (Reviewer ruw5, Reviewer vBwQ)

- **"experiments show SOTA results"** (Reviewer JF8r,  Reviewer ruw5)

- **"implement an INT4 + 2:4 sparse GEMM using NVIDIA CUTLASS"** (Reviewer ruw5,Reviewer vBwQ)

The reviewers also raised insightful and constructive concerns. We made every effort to address all the concerns by providing detailed descriptions and requested results. Here is the summary of the major revisions:

1. **Improve performance on 2:4/4:8 sparsity (Review JF8r):** We have added experiments demonstrating that semi-structured pruning performance can be further improved when using more advanced rotation mechanisms. In addition, we introduce a channel permute technique that significantly boosts the performance of semi-structured pruning to match that of unstructured pruning with negligible overhead.

2. **Experiments on broader benchmarks such as long-context, CoT reasoning  (Reviewer JF8r, Reviewer ruw5, Reviewer vBwQ):** Following the reviewer's question, we add experiments on LongBench for long-context scenarios and math reasoning on the GSM8K benchmark.

3. **Experiments with MXFP4 format (Reviewer JF8r):** Following the reviewer’s suggestion, we replace the common integer quantization with the MXFP4 format and reported the corresponding results.

4. **Results on end-to-end inference speed (Reviewer ruw5):** As per the reviewers' request,  we provide end-to-end efficiency comparisons in terms of TTFT and throughput.

5. **Discussion on time overhead during calibration (Reviewer vBwQ):** Following the reviewer’s suggestion, we add further discussion regarding the computational complexity of the method.

6. **Theoretical grounding and adaptive design for partition ratio \alpha (Reviewer vBwQ):** We mathematically show that when the error follows a uniform distribution,  $\alpha= 0.5$ is optimal. We also present experimental results where we design an adaptive $\alpha$-selection strategy for real-world distribution.

7. **Experiments with partially coupled channel assumption (Reviewer vBwQ):** As suggested by the reviewer, we add ablation experiments where every two adjacent channels are coupled.

8. **Explain the novelty of joint quantization and pruning (Reviewer Y5Wm):** We provide a detailed explanation of how our proposed method differs from prior joint quantization-and-sparsity approaches, both technically and performance-wise.

9. **Explain differences from prior Hessian-based methods (Reviewer Y5Wm):** We include a detailed comparison of the mathematical approximation, performance advantage compared to previous methods.

The valuable suggestions from reviewers are very helpful for us to revise the paper to a better shape. We'd be very happy to answer any further questions.

Looking forward to the reviewer's feedback :D

---

### Meta-Review · Area_Chair_8AfJ · 2026-01-07

**Summary:**

Several reviewers raised concerns about the breadth of the evaluation beyond basic QA-style benchmarks, particularly regarding long-context performance and reasoning/CoT robustness. The authors addressed these points well by expanding the experimental coverage and demonstrating effectiveness across additional tasks. They also responded to questions about practical industry applicability and strengthened the manuscript with more thorough theoretical analysis. However, the reviewer Y5Wm challenge the novelty of the proposed method.

Overall, I recommend acceptance.

**Reviewer Concerns:**

The authors addressed the concerns about the evaluation, the practical deployment in the industry and the theoretical analysis and justification.

**Reviewer Scores:**

The reviewer Y5Wm brings up the major concerns about the novelty but does not response the reply from the reviewers. The other reviewers agree to acceptance.

---

### Decision · Program_Chairs · 2026-01-26

Accept (Poster)